



# A Reference Open-Source Controller for Fixed and Floating Offshore Wind Turbines

Nikhar J. Abbas[1,2], Daniel S. Zalkind[2], Lucy Pao[1], and Alan Wright[2]

[1]University of Colorado Boulder, Boulder, CO, 80309, USA
[2]National Renewable Energy Laboratory, Boulder, CO 80303, USA

**Correspondence:** Nikhar J. Abbas (nikhar.abbas@colorado.edu)

**Abstract.** This paper describes the development of a new reference controller framework for fixed and floating offshore wind turbines that greatly facilitates controller tuning and represents standard industry practices. The reference wind turbine controllers that are most commonly cited in the literature have been developed to work with specific reference wind turbines. Although these controllers have provided standard control functionalities, they are often not easy to modify for use on other turbines, so it has been challenging for researchers to run representative, fully dynamic simulations of other wind turbine designs. The Reference Open-Source Controller (ROSCO) has been developed to provide a modular reference wind turbine controller that represents industry standards and performs comparably to or better than existing reference controllers. The formulation of the ROSCO controller logic and tuning processes is presented in this paper. Control capabilities such as tip-speed ratio tracking generator torque control, minimum pitch saturation, wind speed estimation, and a smoothing algorithm at near-rated operation are included to provide a controller that is comparable to industry standards. A floating offshore wind turbine feedback module is also included to facilitate growing research in the floating offshore arena. All the standard controller implementations and control modules are automatically tuned such that a non-controls engineer or automated optimization routine can easily improve the controller performance. This article provides the framework and theoretical basis for the ROSCO controller modules and generic tuning processes. Simulations of the National Renewable Energy Laboratory (NREL) 5-MW reference wind turbine and International Energy Agency 15-MW reference turbine on the University of Maine semisubmersible platform are analyzed to demonstrate the controller's performance in both fixed and floating configurations, respectively. The simulation results demonstrate ROSCO's peak shaving routine to reduce maximum rotor thrusts by nearly 14% compared to the NREL 5-MW reference wind turbine controller on the land-based turbine and to reduce maximum platform pitch angles by slightly more than 35% when using the platform feedback routine instead of a more traditional low-bandwidth controller.





# 1 Introduction

As wind turbine research has evolved during the past few decades, the need for reference wind turbine controllers has also changed. Traditionally, reference wind turbine controllers have been used for two primary purposes. First, the control systems research community has extensively used reference controllers as a baseline to compare and evaluate more modern and advanced control algorithms (Lackner and van Kuik, 2010; Schlipf et al., 2013). Second, researchers interested in aero-structural dynamics have used reference controllers to enable dynamic simulations in studies concerning other aspects of the wind turbine (Wayman et al., 2006; Sathe et al., 2013). In both applications, a reference controller provides a standardized method by which wind energy researchers can compare and contrast their various turbine designs, aerodynamic models, structural analysis tools, and more.

There has been a lack of reference controllers that can be easily adapted to a wide variety of different wind turbines. The National Renewable Energy Laboratory (NREL) and the Technical University of Denmark (DTU) have published perhaps the most ubiquitous of these reference turbines and respective controllers with the NREL 5-MW (Jonkman et al., 2009) and DTU 10-MW (Bak et al., 2013; Hansen and Henriksen, 2013) turbine models. Generally, for new turbine models, the tuning processes for these turbines' respective controllers necessitate a control systems engineer to generate linear models of the turbine at multiple operating points using aeroelastic simulation solvers to schedule controller gains. At the very least, someone familiar with the NREL 5-MW reference controller tuning process must be able to adequately modify the controller accordingly, as shown in Griffith and Ashwill (2011), where the reference controller is scaled for a rotor with a novel blade design.

Additionally, trends in the wind energy industry are heavily favoring larger and more flexible rotor designs (Veers et al., 2019). As wind turbines have grown and modeling tools have improved and increased in fidelity, design constraints—such as blade tip deflection—have become increasingly important. Without a controller that performs consistently across turbine designs and is representative of the controllers in the field, dynamic simulations cannot be entirely trusted to provide reliable results that can be used for turbine design. The need to run representative dynamic simulations of large flexible turbines necessitates a controller and controller tuning process that can be implemented consistently by the non-controls engineer in a streamlined fashion.

Finally, completely automated optimization tools for medium- to high-fidelity wind turbine designs are becoming well established in research and industry. These tools—such as the Wind-Plant Integrated System Design & Engineering Model (WISDEM®) (Dykes et al., 2014), HawtOpt (Zahle et al., 2015), and Cp-max (Bottasso et al., 2012)—generally include some element of dynamic wind turbine simulation within the optimization loop. Naturally, changes in the wind turbine design often necessitate an update to the controller. An automated controller tuning process and generalized implementation method provide the opportunity for automated control codesign, where the system and controller are designed concurrently (Garcia-Sanz, 2019; Zalkind et al., 2020).

In addition to the need for a generic controller for land-based and fixed-bottom wind turbines, to the authors' knowledge, the availability of a modern, open-source controller with specific logic for floating offshore wind turbines (FOWTs) is not available. Simply reducing the controller bandwidth, as presented by Larsen and Hanson (2007), has been shown to have



disadvantages (Fleming et al., 2014). For the same reasons that a generic controller with automated tuning processes is useful for land-based turbines and rotor design, a specific control implementation for FOWTs has its own utility.

The motivation for the development of the Reference Open-Source Controller (ROSCO) tool chain is threefold. First, continued development, refinement, and expansion of the Delft Research Controller (Mulders and van Wingerden, 2018) provides a wind turbine controller that is consistent with industry-standard control functionalities for use in the research community.

Second, the implementation of generalized tuning procedures through open-source software provides a means for non-controls engineers to conveniently implement and modify a standard wind turbine controller for use in their research applications. Finally, these generalized tuning procedures and controller implementations provide a framework by which systems design optimization tools such as WISDEM can include a controller for time-domain simulations in the optimization loop.

The structure of this manuscript is as follows. In Section 2, we give a high-level overview of the ROSCO tool chain and im-

plementation methods. Here, we also discuss the structure of the ROSCO controller and some requisite theoretical background for the proceeding sections. In Section 3 and Section 4, we discuss the primary generator torque and collective blade pitch controllers, respectively. In Section 5, we provide details on the primary individual "modules" of the controller. For each module detailed in this paper, we provide a review of its purpose, how the generic tuning processes are applied, and in some cases a brief time-domain simulation result to showcase its functionality. Then, in Section 6, we provide power and loads analysis

results for the ROSCO controller on turbines in both land-based and floating configurations. Finally, Section 7 discusses some conclusions and future directions of ROSCO.

## 2   Foundations of the Reference Open-Source Controller

ROSCO was developed to provide a modular control systems architecture with a Fortran-based software structure similar to that of OpenFAST (NREL, 2019), a complete aero-servo-hydro-elastic wind turbine simulation tool developed at NREL. The

initial work and foundation for the ROSCO controller was done by researchers at the Delft University of Technology and presented in (Mulders and van Wingerden, 2018). The standard controller functionalities are designed to perform comparably with existing reference controllers in the literature, such as the NREL 5-MW and DTU 10-MW controllers. In addition to the standard control functionalities, a number of controller features are included that are considered to be similar to those on many industry turbines. The primary functions of the controller are still to maximize power in below-rated operation and to regulate

rotor speed in above-rated operation. The controller is available for download and implementation on GitHub (NREL, 2021a):

<code>https://github.com/NREL/ROSCO.</code>

The controller was developed to communicate with wind turbine simulation software (e.g., OpenFAST) using the Bladed-style control interface (DNV-GL, 2018). The controller source code is compiled once and reads a controller input file. The controller input file can be renamed and changed for any horizontal-axis wind turbine, and it is generally referred to as the

"DISCON.IN" file.

To facilitate usage of ROSCO, a related "ROSCO toolbox" was developed. The ROSCO toolbox is a Python-based tool set developed for tuning, implementing, and post-processing OpenFAST simulations using ROSCO. The ROSCO toolbox is



available for download and implementation on GitHub as well (NREL, 2021b):

$$\texttt{https://github.com/NREL/ROSCO\_toolbox.}$$

The primary purpose of the ROSCO toolbox is to automatically execute the generalized tuning procedures for a given OpenFAST turbine model and to generate the necessary input file for the controller. A separate .yaml (Ben-Kiki et al., 2009) formatted parameter file is used for these generic tuning procedures. This file includes relevant turbine and controller tuning parameters that are generally available a priori. Only four parameters are *necessary* for tuning a complete controller, though additional tuning inputs are available for the user to introduce further modifications and fine-tune the controller performance. In 95 addition to the controller tuning scripts, post-processing scripts, plotting scripts, and a MATLAB/Simulink version (MATLAB, 2019) of ROSCO are available through the ROSCO toolbox.

The general workflow for using the ROSCO tool chain is shown in Figure 1. The tuning file provides necessary controller tuning inputs to the ROSCO toolbox. The ROSCO toolbox then reads an OpenFAST model and writes the DISCON.IN file. The DISCON.IN file is used as the input file to the ROSCO controller and can be directly modified to change the controller 100 performance, to turn on and off individual control modules, or to change the desired control logic. OpenFAST communicates with ROSCO using the bladed-style control interface.

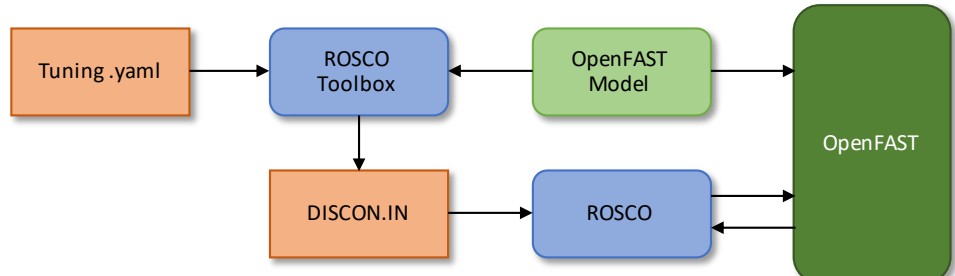

**Figure 1.** The general workflow of the ROSCO tool chain. The orange squares denote ROSCO-related input files, the blue squares denote the ROSCO software tools themselves, and the green squares denote the OpenFAST wind turbine model and OpenFAST itself.

The work by Mulders and van Wingerden (2018) modularized the controller architecture and established an input file framework for the compiled Fortran code. Since then, numerous modules and controller features have been introduced and the functionalities of the controller have been expanded significantly into what is now ROSCO. In this article, we provide an 105 overview of the primary controller modules that are generically tuned by the ROSCO toolbox or were added specifically for ROSCO.

### 2.1 Control Regions

There are generally two methods of actuation in the standard reference wind turbine controllers, including ROSCO. A variable-speed generator torque controller is used to control the generator power, and a collective blade pitch controller is used to 110 regulate rotor speed. These methods of actuation are commonly separated into four regions of operation, with transition logic





between them. The steady-state operating points, as shown in Figure 2 for the NREL 5-MW and International Energy Agency (IEA) 15-MW wind turbines, are a convenient way to visualize the regions of operation.

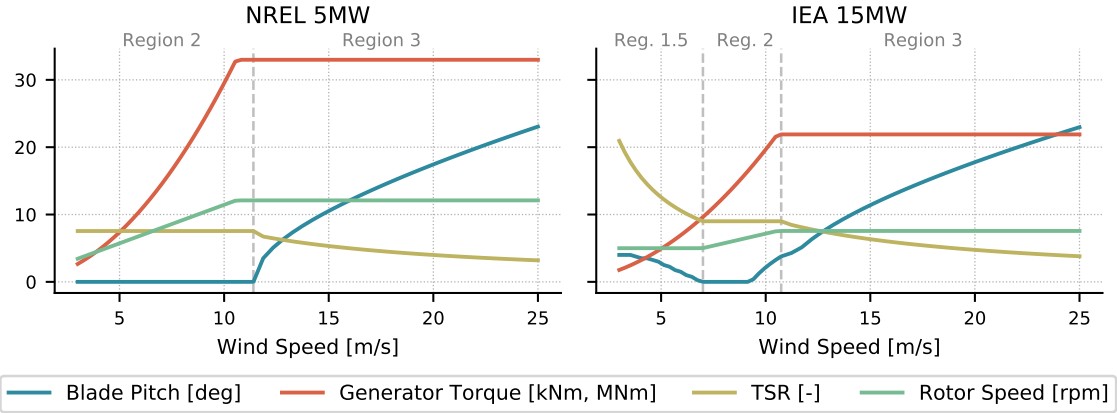

**Figure 2.** The steady-state operating points for the NREL 5-MW and IEA 15-MW wind turbines with the ROSCO controller are shown to illustrate how the turbine operation changes in different regions for different types of controller implementation. The plotted operation points are based on the turbine parameters needed by the ROSCO toolbox tuning procedures and from the $C_p$ surfaces generated by cc-blade (see Section 2.3). The different control regions are separated by the vertical grey dashed lines. Regions 1 and 4 are not shown. The pitch-saturation routine (see Section 5.3) used by the IEA 15-MW wind turbine results in the nonzero blade pitch trajectory in Region 1.5 and across Region 2 and Region 3. Also note that the generator torque is in $kNm$ for the NREL 5-MW turbine and $MNm$ for the IEA 15-MW turbine and that the y-axis is shared for both plots.

In Figure 2, regions 1 and 4 are considered to be below cut-in wind speed and above cut-out wind speed, respectively, so they are not of particular interest. Here, we provide a brief overview of each region and compare how they are implemented in the NREL 5-MW reference wind turbine controller versus ROSCO. In the following sections, we provide more in-depth descriptions of the logic and tuning processes.

**Region 1** is when the wind speed is below the turbine's cut-in wind speed. This region is generally uninteresting for standard control purposes, and it is not shown in Figure 2 for either turbine.

**Region 1.5** is when the wind speed is above the turbine's cut-in wind speed but the turbine cannot operate at its optimal tip-speed ratio (TSR). In the traditional NREL 5-MW reference controller, a linear transition from no generator torque to the minimum optimal generator torque is used. In ROSCO, a proportional-integral (PI) controller modifies the generator torque to maintain a defined minimum rotor speed, and the blades are pitched to their minimum allowable blade pitch angle. In turbines designed to operate at a higher TSR in low wind speeds because of a minimum rotor speed constraint, minimum blade pitch angles can be scheduled by a wind speed estimate to improve power output (see Section 5.3.2). ROSCO's different Region 1.5 behaviors are shown in Figure 2, where Region 1.5 does not span a specific range of wind speeds for the NREL 5-MW wind





turbine because of the use of a PI controller and increased blade pitch angles are seen in low wind speeds for the IEA 15-MW turbine.

**Region 2** is when the wind speed is large enough that the turbine can operate at its optimal TSR but is still below rated. Here, the torque controller aims to maximize power as much as possible, and the blade pitch angle is fixed. In ROSCO, the generator torque controller either follows a traditional squared law (Bossanyi (2000)) or employs a TSR tracking PI controller (see Section 3.1.2) to track a rotor speed reference that is based on a wind speed estimate (see Section 5.1) and optimal TSR. The blades are pitched to fine pitch, where they are generally designed to be the most aerodynamically efficient.

**Region 2.5** is when the wind speed is near rated. Here, the turbine is operating in transition between regions 2 and 3. In the NREL 5-MW reference controller, a linear transition and switching logic are used across a specific range of wind speeds. In ROSCO, a PI controller is used in both the pitch and torque controllers to regulate the rotor speed to their respective set points. A set point smoother (see Section 5.2) is used to ensure that only the pitch or torque controller is active at any point in time and that there is a smooth transition between them. Because of the use of PI controllers and the set point smoother, there is no specific range of wind speeds for Region 2.5 in ROSCO.

**Region 3** is when the wind speed is above rated. The blade pitch controller regulates the rotor speed, and the generator torque is either constant or maintains constant power output. In ROSCO, a gain-scheduled PI collective pitch controller is used to regulate the rotor speed. For floating systems, the NREL 5-MW reference controller reduces the pitch controller bandwidth to prevent platform instability. In ROSCO, an additional feedback term is added to this controller for FOWTs (see Section 5.5) to help stabilize the platform using collective pitch, without the need to detune the rotor speed controller. The generator torque is either kept constant at its rated torque value, or the controller adjusts the torque to maintain constant power output.

**Region 4** is when the wind speed is greater than the turbine's cut-out wind speed. Here, the blades are pitched to reduce rotor thrust to zero. Other than triggering a shutdown maneuver, this region is also generally uninteresting for standard control purposes and is not shown in Figure 2.

In the case of the generator torque and blade pitch controllers, a generator speed signal is low-pass filtered and fed back to the respective controllers. The low-pass filter can be chosen as a first- or second-order filter, and the ROSCO toolbox tuning processes generically tunes the cutoff frequency for the filter to be one-quarter of the blade's first edgewise natural frequency, as suggested in Jonkman et al. (2009). This helps prevent the controller from actuating at a frequency that excites the first edgewise mode of the blade.

## 2.2 Implementation Structure

The high-level block diagram of ROSCO shown in Figure 3 provides an overview of the control modules that are implemented in ROSCO. The combination of these modules ensures smooth actuation across the range of operation regions, as described in Section 2.1. Table 1 provides a brief description of each controller module.





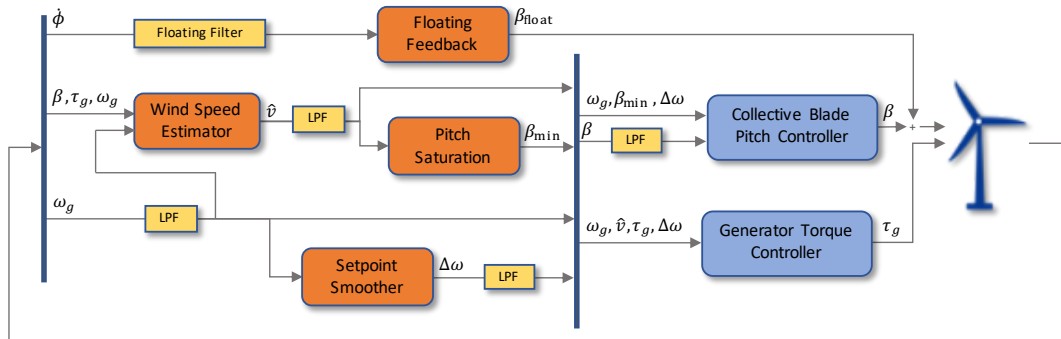

**Figure 3.** A block diagram showing the general controller logic in ROSCO. The generator torque and collective blade pitch controllers described in sections 3 and 4, respectively, are shown in blue. The orange squares denote the various controller modules that are described in Section 5. The yellow squares are filters. The details of the floating filter are described in Section 5.5, and the standard low-pass filter implementations are provided in Appendix A. For the feedback signals, $\phi$ is the tower-top pitch angle, $\beta$ is the collective blade pitch angle, $\omega_g$ is the generator speed, and $\tau_g$ is the generator torque. For outputs from the controller modules, $\beta_{\text{float}}$ is the floating controller's contribution to the blade pitch angle (see Section 5.5), $\beta_{\text{min}}$ is the minimum blade pitch angle (see Section 5.3), $\hat{v}$ is the estimated wind speed (see Section 5.1), and $\Delta\omega$ is a controller reference set point shifting term from the setpoint smoother (see Section 5.2).

## 2.3 The Power Coefficient Surface

Many tuning procedures and module implementations in this work are based on the so-called $C_p$ surface. The power coefficient, $C_p$, is the ratio of the power extracted from the wind to the power available in the wind (Burton et al., 2011). For any given
wind speed, the $C_p$ surface can be calculated to represent the aerodynamic efficiency of the turbine as a function of collective blade pitch angle and TSR. The TSR is the ratio of the speed of the tip of the wind turbine blade to the rotor-averaged wind speed:

$$\lambda = \frac{\omega_r R}{v}, \tag{1}$$

where $\omega_r$ is the rotor speed, $R$ is the rotor radius, and $v$ is the wind speed. Similar parameter surfaces can be generated for the
thrust and torque coefficients of the turbine, $C_t$ and $C_q$, respectively. An example $C_p$ surface is shown in Figure 4 for the IEA 15-MW wind turbine (Gaertner et al., 2020).

The overlaid lines on the $C_p$ surface shown in Figure 4 represent the expected steady-state operation point of the wind turbine. The dash-dotted black line shows the steady-state operating points at low wind speeds, where the TSR is high because of a constraint on the wind turbine's minimum rotor speed. By imposing a blade pitch angle that is greater than fine pitch at
low wind speeds, the turbine can operate at a higher point on the $C_p$ surface. In below-rated operation, the turbine is expected to operate with a fixed blade pitch angle and a constant TSR (see Figure 2); this operation point is denoted by the red dot in Figure 4. In above-rated wind speeds, the blades pitch to regulate the rotor speed, and there is a reduction in TSR (see Figure 2), as denoted by the dashed blue line in Figure 4.





**Table 1.** The primary ROSCO modules and each of their available functionalities. The flag column shows the flag that must be set in the controller input file to activate each functionality. The functionalities column gives the possible flag values and a high-level description of available functions for each module. Section 5 gives more details of these functionalities. Modules with a section number reference in this table are primary modules that are discussed in more detail in this article. Functionalities with an asterisk denote functionalities that are used in the traditional NREL 5-MW wind turbine controller.

| Module | Flag | Functionalities |
|---|---|---|
| Blade pitch controller (Section 4) | `PC_ControlMode` | (0) Fixed pitch (for debug) |
| | | (1)* Gain-scheduled blade pitch control to regulate rotor speed |
| Generator torque controller (Section 3) | `VS_ControlMode` | (0) Square law below rated, constant torque above rated |
| | | (1)* Square law below rated, constant power above rated |
| | | (2) Tip-speed ratio tracking below rated, constant torque above rated |
| | | (3) Tip-speed ratio tracking below rated, constant power above rated |
| Set point smoother (Section 5.2) | `SS_Mode` | (0) No set point smoothing |
| | | (1) Set point smoothing |
| Yaw control | `Y_ControlMode` | (0) No yaw control |
| | | (1) Yaw rate control |
| | | (2) Yaw by individual pitch control |
| Wind speed estimator (Section 5.1) | `WE_Mode` | (0) Filtered hub-height wind speed estimate |
| | | (1) Immersion and invariance estimator |
| | | (2) Extended Kalman filter |
| Individual pitch control | `IPC_ControlMode` | (0) No individual pitch control |
| | | (1) 1P reductions |
| | | (2) 1P and 2P reductions |
| Pitch saturation (Section 5.3) | `PS_Mode` | (0) No pitch saturation |
| | | (1) Pitch saturation |
| Shutdown (Section 5.4) | `SD_Mode` | (0) No shutdown control |
| | | (1) Shutdown at maximum pitch |
| Floating-specific feedback (Section 5.5) | `Fl_Mode` | (0) No floating-specific feedback |
| | | (1) Include floating feedback term |
| Distributed aerodynamic control | `Flp_Mode` | (0) No flap control |
| | | (1) Constant flap angle |
| | | (2) PI flap control |

In the standard ROSCO toolbox tuning procedure, the steady-state blade element momentum solver cc-blade (Ning, 2013) is used to find the $C_p$, $C_t$, and $C_q$ surfaces quickly and efficiently. Similarly, parameter surfaces can be generated and written to a text file using any other blade element momentum or coupled aeroelastic solver. Using more complex solvers will, naturally,


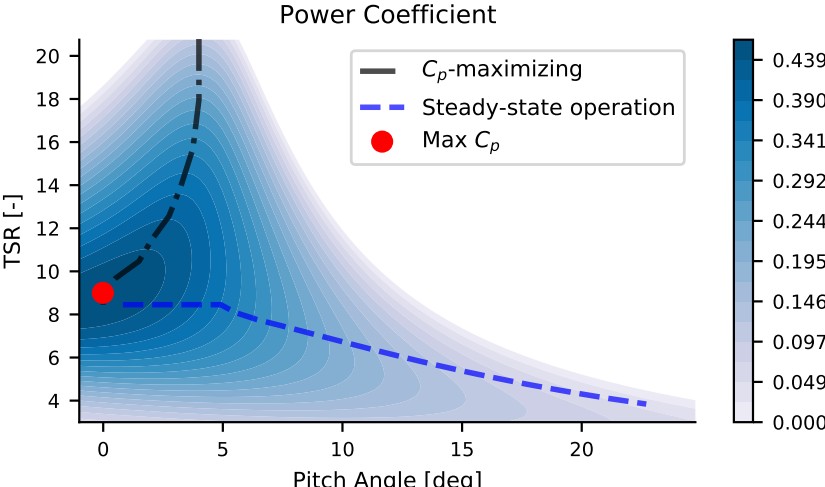

**Figure 4.** $C_p$ surface for the IEA 15-MW wind turbine. The dash-dotted black line shows the below-rated steady-state operating points that result from a $C_p$-maximizing minimum pitch schedule at low wind speeds when the turbine has a minimum rotor speed constraint. The dashed blue line shows the expected steady-state operation points that are used to calculate the controller gain schedules.

provide a more realistic $C_p$ surface, though at a significant increase in computation time. Anecdotally, the authors have found the $C_p$-surfaces generated using cc-blade to be sufficiently accurate for controller synthesis on numerous wind turbines thus far. Future work includes a more substantive analysis of the controller performance sensitivity to $C_p$-surfaces generated using

different aerodynamic solvers.

Within the ROSCO toolbox tuning processes, the $C_p$ surface is primarily used to facilitate the calculation of the plant parameters, as discussed in Section 2.4. These parameters are used for tuning the blade pitch and generator torque controller gains and for accurate implementation of the wind speed estimator. The $C_p$ surface is also used to determine the minimum blade pitch schedule in low wind speeds, if needed, and to determine the optimal below-rated TSR to maximize power output.

Finally, the $C_p$ surface is used within ROSCO itself so that the wind speed estimator can accurately estimate the operational state of the wind turbine. The $C_t$ surface is used to determine the minimum blade pitch schedule for the so-called "peak shaving" routine, as described in Section 5.3.1.

### 2.4 Plant Model

A number of generic tuning processes have been developed for the the ROSCO controller. The first publication on this work

(Abbas et al., 2020) provided an initial formulation of the primary tuning processes in ROSCO. By using simplified analytical models of the wind turbine, the need for numerical linearization routines using servo-aeroelastic codes such as OpenFAST is avoided. To define the simplified wind turbine for control systems development, a first-order model of the wind turbine is used



(Pao and Johnson, 2011):

$$\dot{\omega}_g = \frac{N_g}{J}(\tau_a - N_g \tau_g) \tag{2}$$

where the aerodynamic torque is:

$$\tau_a = \frac{1}{2}\rho A_r \frac{C_p(\lambda,\beta)}{\omega_r}v^3. \tag{3}$$

In (2), $\omega_g$ is the generator speed, $J$ is the rotor inertia, $N_g$ is the gearbox ratio, and $\tau_g$ is the generator torque. In (3), $\rho$ is the air density, and $A_r$ is the rotor area. The power coefficient $C_p(\lambda,\beta)$ is found via lookup table. The first-order linearization of (3) at some nominal steady-state operational point is:

$d\tau_a = \Gamma_{\omega_g}\big|_{\text{op}} d\omega_g + \Gamma_\beta\big|_{\text{op}} d\beta + \Gamma_v\big|_{\text{op}} dv, \tag{4}$

where $\Gamma_{\omega_g} = \frac{\partial \tau_a}{\partial \omega_g}$, $\Gamma_\beta = \frac{\partial \tau_a}{\partial \beta}$, and $\Gamma_v = \frac{\partial \tau_a}{\partial v}$; and "op" denotes the expected, steady-state operational $\omega_g$, $\beta$, and $v$ for any linearization point. Equation (2) can then be rewritten in a linearized form as:

$$d\dot{\omega}_g = A(v_{\text{op}})d\omega_g + B_{\tau_g}d\tau_g + B_\beta(v_{\text{op}})d\beta + B_v(v_{\text{op}})dv, \tag{5}$$

Equation (5) is the plant model used to define the generator torque and blade pitch controller gains, where:

$A(v_{\text{op}}) = \frac{1}{J}\frac{d\tau_a}{d\lambda}\frac{d\lambda}{d\omega_g} \tag{6}$

$$\frac{d\tau_a}{d\lambda} = \frac{N_g}{2}\rho A_r R v^2 \frac{1}{\lambda_{\text{op}}^2}\left(\frac{dC_p}{d\lambda}\lambda_{\text{op}} - C_{p,\text{op}}\right) \tag{7}$$

$$\frac{d\lambda}{d\omega_g} = \frac{1}{N_g}\frac{R}{v_{\text{op}}} \tag{8}$$

The coefficients $B_{\tau_g}$ and $B_\beta(v_{\text{op}})$ are described in sections 3.1.2 and 4, where the blade pitch and generator torque controllers are described, respectively. To calculate the $C_p$ surface gradients, a second-order central differencing approach is used.

Notably, this linearized plant model is used to separately tune above- and below-rated controllers. In above-rated operation, the generator torque is assumed to be constant, so $d\tau_g = 0$. Similarly, in below-rated operation, the blade pitch angle is assumed to be constant, so $d\beta = 0$. Finally, $B_v$, the disturbance ($dv$) input matrix to the system, is set to 0 for the controller tuning.

## 2.5  Proportional-Integral Reference Tracking Control

A reference tracking PI controller is used in above-rated and below-rated operation. The PI controller is generically defined as:

$y = k_p u + k_i \displaystyle\int_0^T u\,dt, \tag{9}$

where $u$ is an input to the controller; $y$ is the output from the controller passed to the wind turbine; and $k_p$ and $k_i$ are the proportional and integral gains, respectively. For example, the standard PI blade pitch controller has inputs and outputs such





that:

$$u = d\omega_g, \qquad y = d\beta, \tag{10}$$

and the TSR tracking PI generator torque controller has inputs such that:

$$u = d\omega_g, \qquad y = d\tau_g. \tag{11}$$

Combining the PI controller formulation in (9) with the linear model in (5) in a standard negative-feedback loop results in the closed-loop transfer function:

$$H(s) = \frac{d\Omega_g(s)}{d\Omega_{g,\text{ref}}(s)} = \frac{B(k_p(v_{\text{op}})s + k_i(v_{\text{op}}))}{s^2 + (Bk_p(v_{\text{op}}) - A(v_{\text{op}}))s + Bk_i(v_{\text{op}})}, \tag{12}$$

where the subscript "$g,\text{ref}$" denotes the generator speed reference speed; and $B$ is either $B_{\tau_g}$ or $B_\beta(v_{\text{op}})$, depending on whether the turbine is in below- or above-rated operation, respectively. For a standard horizontal-axis wind turbine, $B$ is negative in below- and above-rated operation, so $k_i$ and $k_p$ are generally negative.

This closed-loop system is a simple second-order system, so we define the PI gains as:

$$k_p(v_{\text{op}}) = \frac{1}{B}(2\zeta_{\text{des}}\omega_{\text{des}} + A(v_{op})), \tag{13}$$

$$k_i(v_{\text{op}}) = \frac{\omega_{\text{des}}^2}{B}, \tag{14}$$

where the closed-loop response is characterized by a desired natural frequency, $\omega_{\text{des}}$, and damping ratio, $\zeta_{\text{des}}$. By defining the desired natural frequency and damping ratio of the rotor speed response, the user can modify the dynamic response of the rotor to changes in the wind speed. Increased desired natural frequencies will decrease the response time of the rotor, whereas increased damping ratios will reduce the amount of oscillation in the response. To learn more about second-order system
responses, the authors recommend Franklin et al. (2019) to the curious reader.

If PI controllers are employed for below- and above-rated operation, we are left with the need to define four controller tuning inputs. The choice of $\omega_{\text{des}}$ and $\zeta_{\text{des}}$ for the below- and above-rated operating regions are the only parameter decisions that *need* to be made by the control designer or optimization routine, though additional tuning of the individual modules discussed in Section 5 might further improve controller performance.

**2.6   Filters**

Four filters are commonly used in the ROSCO controller (see Figure 3): first- and second-order low-pass filters, a first-order high-pass filter, and a notch filter. Appendix A shows the continuous-time formulations of these filters. The filters are converted to discrete time for implementation using the bilinear transform.





## 3    The Generator Torque Controller

Four total variable-speed generator torque control methods are available in ROSCO (see Table 1). Two methods of optimal

power generation are available for below-rated operation, and two methods of maintaining power output near the turbine's

rated value are available in above-rated operation.

### 3.1    Below-Rated Torque Control

In below-rated operation, the generator torque controller's goal is to maximize power production. If the blades are pitched to

fine pitch, maximum aerodynamic efficiency can be achieved and power can be maximized if the torque controller maintains

the TSR corresponding to the peak of the $C_p$ surface. In ROSCO, this is done in one of two ways.

#### 3.1.1    $K\omega^2$ Law

A study of equation (2) in steady state, such that $\dot{\omega}_g = 0$, provides the foundation for the so-called "$K\omega_g^2$ law" for optimal torque

control (Bossanyi, 2000; Johnson et al., 2006). By restructuring equation (2), and assuming that the wind turbine is operating

at its optimal TSR, $\lambda_{\mathrm{opt}}$, and corresponding power coefficient, $C_{p,\mathrm{max}}$, one can realize a quadratic relationship between the

generator speed and demanded generator torque. This relationship is commonly defined as:

$$\tau_g(t) = K\omega_g^2(t), \quad \text{where} \quad K = \frac{\pi\rho R^5 C_{p,\mathrm{max}}}{2\lambda_{\mathrm{opt}}^3 N_g^3 \eta_{\mathrm{gen}}\eta_{\mathrm{gb}}}. \tag{15}$$

In equation (15), $\eta$ denotes efficiencies of the generator (gen) and gearbox (gb).

#### 3.1.2    Tip-Speed Ratio Tracking Torque Control

Two primary motivations are behind the development of the TSR tracking controller in ROSCO. First, although the $K\omega_g^2$

law has historically worked reliably in idealized simulation environments, the calculation of $K$ can be subject to modeling

and assumption errors. For example, the assumption that $R$ is a constant value across wind speeds does not hold as strongly

in modern, highly flexible rotors as it has in the past. Fortunately, modern rotors are still commonly designed to maximize

aerodynamic efficiency at a specific TSR. Second, industry collaborators have indicated that a TSR tracking controller is more

representative of the controllers often used in the field.

If the blades are pitched to their most aerodynamically efficient angle and the wind turbine rotor is operating at its optimal

TSR, the turbine power is theoretically maximized. This suggests that if the wind speed can be measured or estimated accu-

rately, a generator torque controller can be designed to maintain the rotor's $\lambda_{\mathrm{opt}}$ and maximize power capture. In ROSCO, a

standard PI controller is used to track a generator speed reference. For the generator torque controller:

$$\omega_{\mathrm{ref},\tau} = N_g \frac{\lambda_{\mathrm{opt}}\hat{v}}{R}, \tag{16}$$





where $\hat{v}$ is the estimated rotor-effective wind speed provided by the wind speed estimator described in Section 5.1. This reference signal is also constrained by:

$$\omega_{g,\min} \leq \omega_{\mathrm{ref},\tau} \leq \omega_{g,\mathrm{rated}}, \tag{17}$$

where $\omega_{g,\min}$ is the minimum allowable generator speed.

A straightforward and automated tuning process has been developed for the PI gains for the TSR tracking torque controller. With the assumption that the blade pitch is held constant at fine pitch in below-rated operation, $d\beta = 0$ in (5). This means that $B = B_{\tau_g}$ in equations (13) and (14) and is defined as:

$$B = B_{\tau_g} = \frac{-N_g^2}{J} \tag{18}$$

for the generator torque controller. With this, $\omega_{\mathrm{des}}$ and $\zeta_{\mathrm{des}}$ can be chosen to describe the below-rated closed-loop rotor speed response, and (13) and (14) can be calculated for the generator torque controller. Equation (13) suggests that $k_p(v)$ and $A(v)$ are both dependent on $v$; however, it was found that defining $k_{p,\mathrm{vs}} = k_p(v = v_{\mathrm{rated}})$, where the subscript "vs" denotes variable-speed torque control, provides less erratic control actuation without negatively affecting the power capture.

### 3.2 Above-Rated Torque Control

There are two standard methods of actuating the generator torque in above-rated operation. Defining $P$ as the generator power output, in above-rated operation the generator torque is defined to be:

$$\tau_{g,\mathrm{ar}}(t) = \frac{P_{\mathrm{rated}}}{\omega_{g,\mathrm{rated}}}, \quad \text{or} \quad \tau_{g,\mathrm{ar}}(t) = \frac{P_{\mathrm{rated}}}{\omega_g(t)}, \tag{19}$$

for constant torque or constant power output, respectively, where the subscript "rated" denotes the value as calculated at rated operation.

If the $K\omega_g^2$ law is used for below-rated torque control, the torque controller switches to above-rated operation when the blades are pitched beyond an offset, denoted by `PC_Switch` in the DISCON.IN file. When the turbine is in above-rated operation, the generator torque is then defined directly by the relationships in (19). If the TSR tracking control is used for below-rated operation, the above-rated generator torque output is simply constrained such that $\tau_g(t) \leq \tau_{g,\mathrm{ar}}(t)$. In above-rated operation, the set point smoother (see Section 5.2) shifts the reference generator speed such that the generator torque is saturated at its maximum allowable value, resulting in either constant torque or constant power output.

If constant torque control is used, power output changes are directly correlated to the changes in the generator speed in above-rated operation. If constant power control is used, there are still some changes in power output because $\omega_g(t)$ in (19) is low-pass filtered, but the power is much more consistent. As discussed by Jonkman (2010), using a constant generator torque controller in above-rated operating conditions can help improve FOWT platform stability.



## 4 The Blade Pitch Controller

In below-rated operation, the generator speed is less than the rated generator speed, so the blade pitch angle, $\beta$, saturates at $\beta = \beta_{\min}$. This is generally the fine-pitch angle, unless a pitch saturation schedule is implemented, as described in Section 5.3.

In above-rated operation, a PI controller is used to determine the collective blade pitch angle to keep the turbine at a rotor speed reference. The above-rated rotor speed reference $\omega_{g,\text{ref}}$ for the blade pitch controller is generally defined as

$$\omega_{\text{ref},\beta} = \omega_{\text{rat}}. \tag{20}$$

It is well established, and common in reference controllers, to employ a gain schedule to improve blade pitch controller performance (Jonkman et al., 2009; Hansen and Henriksen, 2013; Mulders and van Wingerden, 2018). In the ROSCO toolbox, we use the $C_p$ surface to prescribe this gain schedule, rather than using aero-servo-elastic linearization tools.

When tuning the pitch controller's gain schedule, it is assumed that the turbine's generator torque is kept constant, so $d\tau_g = 0$ in (5). Consequently, $B = B_\beta(v_{\text{op}})$ in equations (13) and (14) is:

$$B = B_\beta(v_{\text{op}}) = \frac{N_g}{J}\frac{\partial \tau_a}{\partial \beta} = \frac{N_g}{2J}\rho A_r R v_{\text{op}}^2 \frac{1}{\lambda_{\text{op}}^2}\left(\left.\frac{\partial C_p}{\partial \beta}\right|_{\lambda_{\text{op}},\beta_{\text{op}}}\lambda_{\text{op}}\right). \tag{21}$$

For any wind speed $v_{\text{op}}$, the expected steady-state $C_{p,\text{op}}$ can be calculated by (Bottasso and Croce, 2009):

$$C_{p,\text{op}} = C_{p,\text{rated}}\left(\frac{\lambda_{\text{op}}}{\lambda_{\text{rated}}}\right)^3, \tag{22}$$

Through the relationship in (22) and the $C_p$ surface, we can find the expected blade pitch angles, $\beta_{\text{op}}(\lambda_{\text{op}})$, for any TSR. In steady-state above-rated operation, the generator speed is considered to be constant, so the TSR is only a function of the wind

speed, and we can define $\beta_{\text{op}}(v)$. This enables us to change $A(v_{\text{op}})$ and $B_\beta(v_{\text{op}})$ to be $A(\beta_{\text{op}})$ and $B_\beta(\beta_{\text{op}})$ for controller tuning purposes. Equations (13) and (14) are then defined for the blade pitch controller to be $k_{p,\text{pc}}(\beta_{\text{op}})$ and $k_{i,\text{pc}}(\beta_{\text{op}})$, and the blade pitch controller gain schedule can be implemented as a function of blade pitch angle rather than an estimated wind speed. In the ROSCO implementation of the blade pitch controller, a low-pass-filtered blade pitch angle, $\beta_{\text{lpf}}$, is used to interpolate the necessary pitch controller gains, $k_{p,\text{pc}}(\beta_{\text{lpf}}(t))$ and $k_{i,\text{pc}}(\beta_{\text{lpf}}(t))$, for the PI controller at each time step.

Notably, the $C_p$ surface gradients used to calculate both $A(v_{\text{op}})$ and $B(v_{\text{op}})$ approach zero near the "peak" of the $C_p$ surface. Because $B_\beta(v_{\text{op}})$ is in the denominator of (13) and (14), the gains would theoretically approach infinity near rated operation. To avoid unrealistically high gains, $A(v_{\text{op}})$ and $B(v_{\text{op}})$ are each approximated by a linear fit for the blade pitch controller gain calculation in the ROSCO toolbox. With this foundation, the blade pitch controller's gain schedule can be generated using the ROSCO toolbox for any user-defined $\omega_{\text{des}}$ and $\zeta_{\text{des}}$.

## 325 5 Additional Control Modules

ROSCO is modularized such that various control methods can be switched on and off without the need to recompile any code. Here, we present the theoretical foundations for the module implementations and their respective generic tuning processes.





### 5.1 Wind Speed Estimator

In this section, we discuss the wind speed estimator used in ROSCO. The wind speed estimate is used in the TSR tracking
generator torque controller (see Section 3.1.2) and pitch saturation (see Section 5.3) routines. The employed wind speed
estimator is inspired by Knudsen et al. (2011) and is based on a continuous-discrete extended Kalman filter. The theoretical
background of the continuous and discrete-time extended Kalman filters used in this work is further detailed in Grewal (2011).
The Kalman filter uses informed definitions of the covariance matrices based on the expected wind fields to provide a wind
speed estimate, $\hat{v}$. The derivatives are evaluated in continuous time, whereas the measurement updates are evaluated in discrete
time. A forward Euler integration method is used to propagate the state and covariance estimates forward in time.

The nonlinear continuous-time state-space model used for the continuous-discrete Kalman filter implemented in the controller is defined as

$$\dot{\boldsymbol{x}} = \boldsymbol{f}(\boldsymbol{x}, \boldsymbol{u}) + \boldsymbol{\xi}_s \tag{23}$$

$$y = h(\boldsymbol{x}, \boldsymbol{u}) + \xi_m, \tag{24}$$

where the system noise is $\boldsymbol{\xi_s} = [n_1 \ n_2 \ n_3]^T$, and $n_i$ is considered to be zero-mean white noise. The measurement noise, $\xi_m$,
is assumed to be white noise with a constant covariance. The system state, inputs, and outputs are defined as:

$$\boldsymbol{x} = [\omega_r \ v_t \ v_m]^T \tag{25}$$

$$\boldsymbol{u} = [\beta \ \tau_g]^T \tag{26}$$

where $v_t$ is the turbulent component of the wind speed, and $v_m$ is the mean wind speed. The nonlinear state equations are
defined as:

$$\dot{\omega}_r = \frac{1}{J}(\tau_a - N_g \tau_g) \tag{27}$$

$$\dot{v}_t = -a(v_m)v_t + n_1 \tag{28}$$

$$\dot{v}_m = n_2 \tag{29}$$

and the output is:

$$y = \omega_r + \xi_m, \tag{30}$$

To complete the state equation definitions, we establish:

$$a(v_m) = \frac{\pi v_m}{2L} \tag{31}$$

$$\hat{v} = v_t + v_m. \tag{32}$$

In (31), $L$ is a turbulence length-scale parameter generically defined as $L = 3D$, where $D$ is the rotor diameter.



In the Kalman filter, the covariance matrices are based on the wind model as defined by Knudsen et al. (2011). In the ROSCO implementation of this wind speed estimator, the process noise ($Q$) and measurement noise ($R_m$) covariances are:

$$Q = \begin{bmatrix} 1e^-5 & 0 & 0 \\ 0 & \frac{\pi v_m^3 t_i^2}{L} & 0 \\ 0 & 0 & \frac{2^2}{600} \end{bmatrix}, \quad R_m = 0.02, \tag{33}$$

where the turbulence intensity is defined as $t_i = 0.18$, the upper limit of the turbulence intensity for standard inflow wind conditions as defined by DNV-GL (2018). A continuous-discrete Kalman filter is then implemented through the following

formulation:

– Prediction update:

$$\dot{\hat{x}}(t) = f(\hat{x}_{k-1|k-1}, u_k), \qquad \text{(Predicted state estimate)} \tag{34}$$

$$\dot{P}(t) = F(t)P_{k|k} + P_{k|k}F^T(t) + Q_k - K_{k-1}R_m K_{k-1}^T, \qquad \text{(Predicted covariance estimate)} \tag{35}$$

– Measurement update:

$$\tilde{y}_k = y_k - h_k(\hat{x}_{k|k-1}), \qquad \text{(Innovation residual)} \tag{36}$$

$$S_k = H_k P_{k|k-1} H_k^T + R_m, \qquad \text{(Innovation covariance)} \tag{37}$$

$$K_k = P_{k|k-1} H_k^T S_k^{-1}, \qquad \text{(Near-optimal Kalman gain)} \tag{38}$$

$$\hat{x}_{k|k} = \hat{x}_{k|k-1} + K_k \tilde{y}_k, \qquad \text{(Updated state estimate)} \tag{39}$$

$$\hat{v}_k = \begin{bmatrix} 0 & 1 & 1 \end{bmatrix} \hat{x}_{k|k}, \qquad \text{(Wind speed estimate)} \tag{40}$$

$$P_{k|k} = (I - K_k H_k) P_{k|k-1}, \qquad \text{(Updated covariance estimate)} \tag{41}$$

where the state transition and output Jacobians are defined as:

$$F(t) = \left. \frac{\partial f}{\partial x} \right|_{\hat{x}_{k-1|k-1}, u_k}, \qquad H_k = \left. \frac{\partial h}{\partial x} \right|_{\hat{x}_{k|k-1}}. \tag{42}$$

Wind speed estimator results from a 10-minute simulation of the NREL 5-MW turbine are shown in Figure 5. In the presented simulation, the root-mean-square error between the rotor-averaged wind speed and the wind speed estimate is 0.48 $m/s$.

**5.2 Set Point Smoothing**

The generator torque and blade pitch controllers will conflict with each other in near-rated operation if the generator torque and blade pitch reference speeds are defined only by (16) and (20). To avoid this, we employ a set point smoother regime that is akin to a Region 2.5 controller (Schlipf; Zalkind and Pao, 2019). Practically, the so-called "set point smoother" shifts the generator speed reference signal of the saturated controller while the unsaturated controller is active, so the controllers do not

have conflicting behaviors. This encourages one controller to stay active while the other is not.




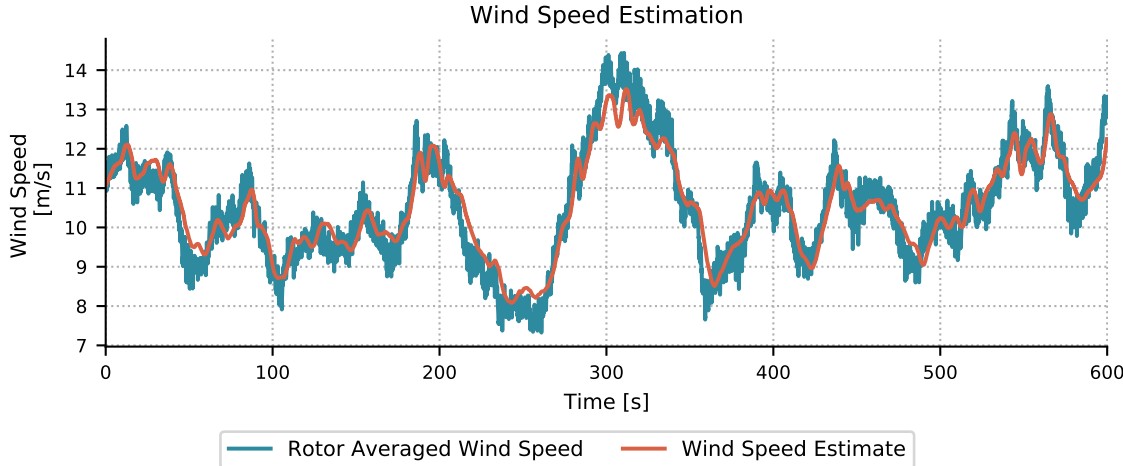

**Figure 5.** Wind speed estimator results for the NREL 5-MW wind turbine with 10 minutes of simulation time and an average wind speed of 11 $m/s$.

We first define an offset to the rotor speed set point, $\Delta\omega$, as:

$$\Delta\omega = \left[ \underbrace{\left(\frac{\beta - \beta_{\min}}{\beta_{\max}}\right) k_{\mathrm{vs}}}_{\Delta\beta} - \underbrace{\left(\frac{\tau_{g,\max} - \tau_g}{\tau_{g,\max}}\right) k_{\mathrm{pc}}}_{\Delta\tau} \right] \omega_{g,\mathrm{rated}}, \qquad (43)$$

where $k_{\mathrm{vs}}$ and $k_{\mathrm{pc}}$ are unitless gain factors that are greater than 0, and $\beta_{\max}$ is the blade pitch angle at cut-out wind speed. Equation (43) is defined such that $\Delta\beta = 0$ in below-rated operation, and $\Delta\tau = 0$ in above-rated operation. A piecewise logic is then implemented to shift the blade pitch or generator torque reference generator speeds:

$$\omega_{\mathrm{ref},\tau} = \begin{cases} \omega_{\mathrm{ref},\tau} - \Delta\omega & \Delta\omega \geq 0 \\ \omega_{\mathrm{ref},\tau} & \Delta\omega < 0 \end{cases} \quad \text{and} \quad \omega_{\mathrm{ref},\beta} = \begin{cases} \omega_{\mathrm{ref},\beta} & \Delta\omega \geq 0 \\ \omega_{\mathrm{ref},\beta} - \Delta\omega & \Delta\omega < 0 \end{cases}. \qquad (44)$$

Figure 6 shows a block diagram displaying the set point smoother logic.

The shifting term in the set point smoother defined in (43) includes normalization terms, so no specific tuning is necessary. The ROSCO toolbox tuning processes define $k_{\mathrm{vs}} = 1$ and $k_{\mathrm{pc}} = 1e-3$. These values were chosen because of their utility across turbine models, although specific tuning of these can improve the controller performance near rated operation.

Figure 7 shows results from a 10-minute time-domain simulations of the NREL 5-MW wind turbine near rated operation. The set point smoother employed in ROSCO provides significantly smoother transitions from above- to below-rated operation. When the blade pitch is greater than zero (the first 12 seconds in Figure 7), the torque controller set point decreases, which biases the generator torque toward its maximum value. When the generator torque is less than its rated value (the last 12 seconds in Figure 7), the set point for the pitch controller increases, and it biases the pitch to its minimum value. The smoother





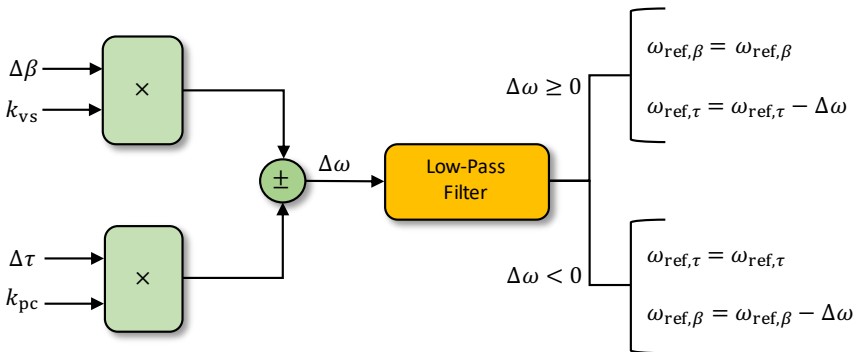

**Figure 6.** A block diagram of the set point smoother logic defined in (43)–(44) is shown here. The term $\Delta\omega$ shifts the blade pitch or generator torque controller to help avoid unwanted controller interactions.

shifts the saturated controller's set point linearly depending on how "far" it is from rated operation. By separating the control regions this way, resonances such as those seen in the time history of the NREL 5-MW torque controller's signal between 15 and 25 seconds can be reduced.

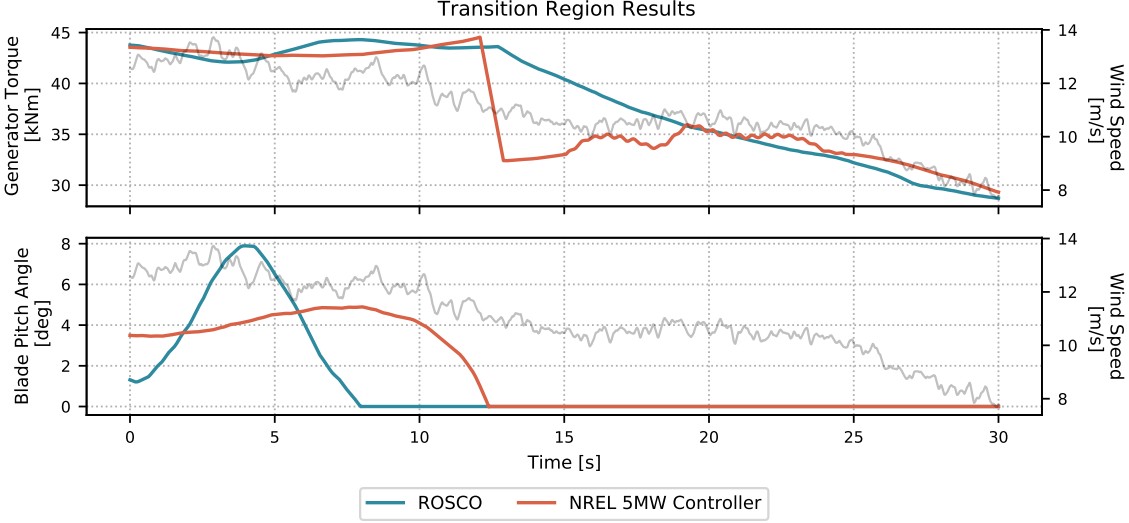

**Figure 7.** Time-domain simulation results from the NREL 5-MW wind turbine near rated wind speeds. The rotor-averaged wind speed is plotted in the background as grey curves in each plot.

## 5.3 Minimum Pitch Saturation

A method for saturating the minimum blade pitch angle for a given wind speed estimate is also included in ROSCO. This has primarily been used for two purposes: to limit the rotor thrust through a peak shaving algorithm and to implement a minimum





pitch angle at low wind speeds for power maximization in the presence of a minimum rotor speed constraint. The ROSCO controller simply defines a minimum blade pitch angle for a given wind speed as defined by a lookup table in the DISCON.IN file. Figure 8 provides an example minimum pitch schedule for the IEA 15-MW wind turbine with a minimum rotor speed constraint and peak shaving, along with the corresponding rotor thrust that is expected with and without peak shaving. The

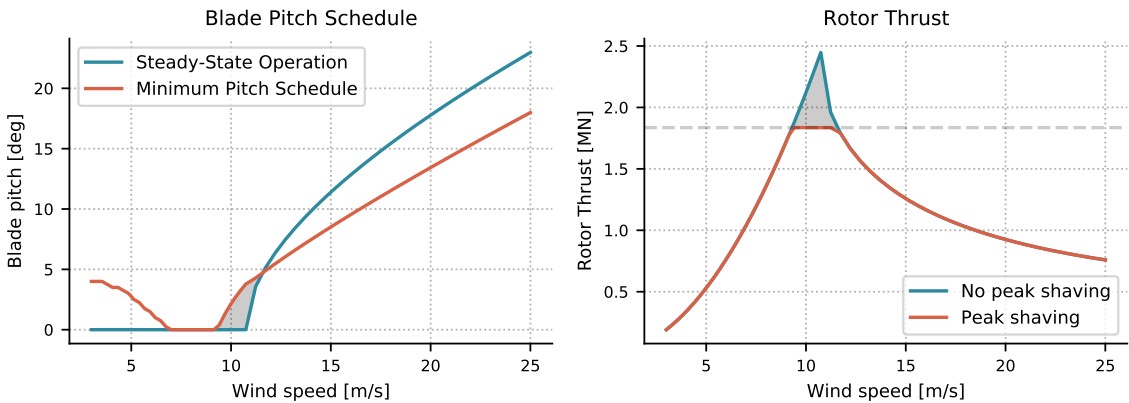

**Figure 8.** Expected steady-state pitch saturation for the IEA 15-MW wind turbine, based on the $C_t$ surface calculated with cc-blade. The blade pitch schedule (left) shows the expected steady-state value of the pitch angle along with the imposed minimum pitch angles with pitch saturation for both peak shaving and power maximization in low wind speeds. The righthand plot shows the expected rotor thrust with and without peak shaving. The grey region in the blade pitch schedule corresponds to the pitch angles that result in a rotor thrust above the maximum allowable thrust, shown as the grey region in the righthand plot.


following two subsections describe the two primary aspects of the minimum pitch schedule in more detail.

### 5.3.1 Peak Shaving

Thrust limiting, or peak shaving, is often used to reduce peak tower base loads. Generally, the largest rotor thrusts are seen near rated operation and have a strong affect on tower base loads. It is has also been shown that rotor thrust is correlated to the
blade pitch actuation (Bossanyi, 2003; Fischer and Shan, 2013; Petrović and Bottasso, 2017). We impose a minimum blade pitch angle, $\beta_{\min}(v)$, to "shave" the peak of the rotor thrust curve and subsequently reduce tower base loads near rated. The rotor thrust can be defined by:

$$T_r(v) = \frac{1}{2}\rho A_r v^2 C_t(\lambda, \beta),\tag{45}$$

where $C_t$ is the rotor thrust coefficient. Given a maximum allowable rotor thrust, $T_{r,\max} = a\max(T_r(v))$, where $a \leq 1$, the
maximum allowable thrust coefficient is defined as:

$$C_{t,\max}(v) = \frac{2T_{r,\max}}{\rho A_r v^2}.\tag{46}$$

For all operational TSRs, we can find a blade pitch angle, $\beta_{\min}(C_{t,\max}, \lambda)$, through a $C_t$ surface similar to the $C_p$ surface shown in Figure 4. Notably, $C_{t,\max}(v)$ and $\lambda$ are both parameterized by $v$, so we can define a minimum blade pitch schedule, $\beta_{\min}(v)$, that is dependent on a wind speed estimate. Figure 9 shows an example 60-second time series for the NREL 5-MW wind

turbine in near-rated operation. Based on the wind speed estimate, the blades are pitched to reduce the rotor thrust to stay at or below the maximum allowable thrust limit.

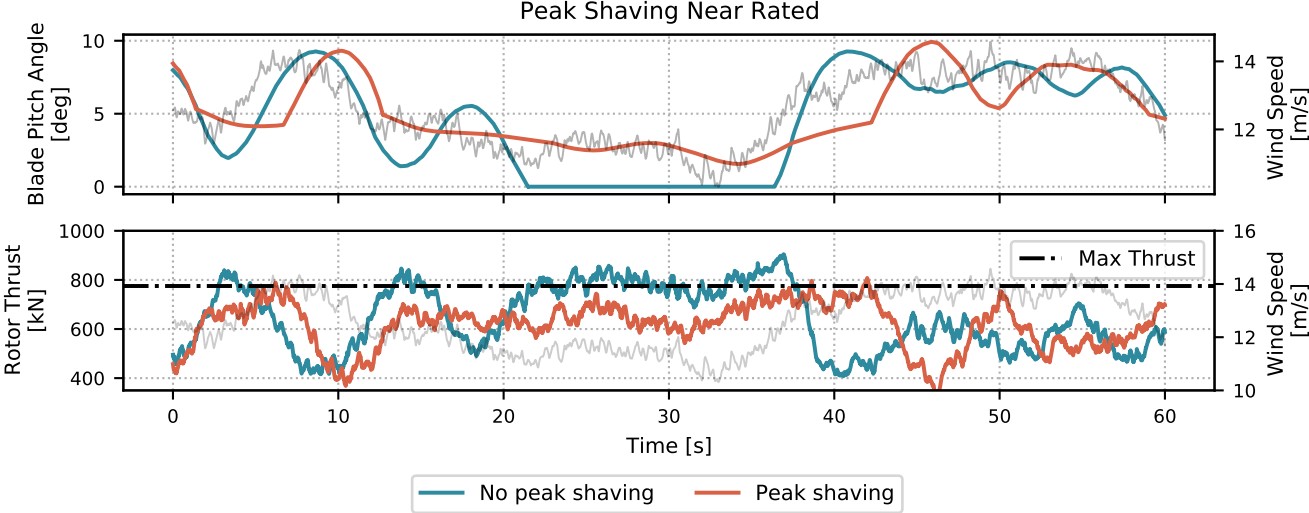

**Figure 9.** A sample 1-minute time series from a 10-minute turbulent simulation of the NREL 5-MW wind turbine to showcase the peak shaving routine in ROSCO. The peak shaving percentage was tuned to 20% for this simulation. The wind speed is shown in the background of each plot, and the maximum allowable thrust is denoted by the dash-dotted black line in the lower plot.

Peak shaving algorithms implemented with minimum pitch saturation can result in power losses because the turbine will no longer be operating at its highest efficiency near rated. Though the default peak shaving algorithm in the ROSCO toolbox reduces the maximum rotor thrust by 25%, ultimately the trade-off between power production and load reductions must be

made by the control system designer.

### 5.3.2 Power Maximization in Low Wind

In certain wind turbine configurations, such as the IEA 15-MW wind turbine (Gaertner et al., 2020), minimum or maximum blade tip-speed limits might be imposed for reasons such as tower resonance avoidance or noise avoidance. If a minimum tip-speed constraint exists, the wind turbine cannot operate at $\lambda_{\text{opt}}$ in low wind speeds. A minimum pitch schedule can be

implemented such that the power can be maximized in low wind speeds while the generator torque controller works to satisfy the minimum rotor speed constraints. The black dash-dotted line in Figure 4 provides insight into how the power output can be improved through a minimum pitch angle at low wind speeds by moving the turbine's expected operating point to the "top" of the $C_p$ surface for high TSRs.



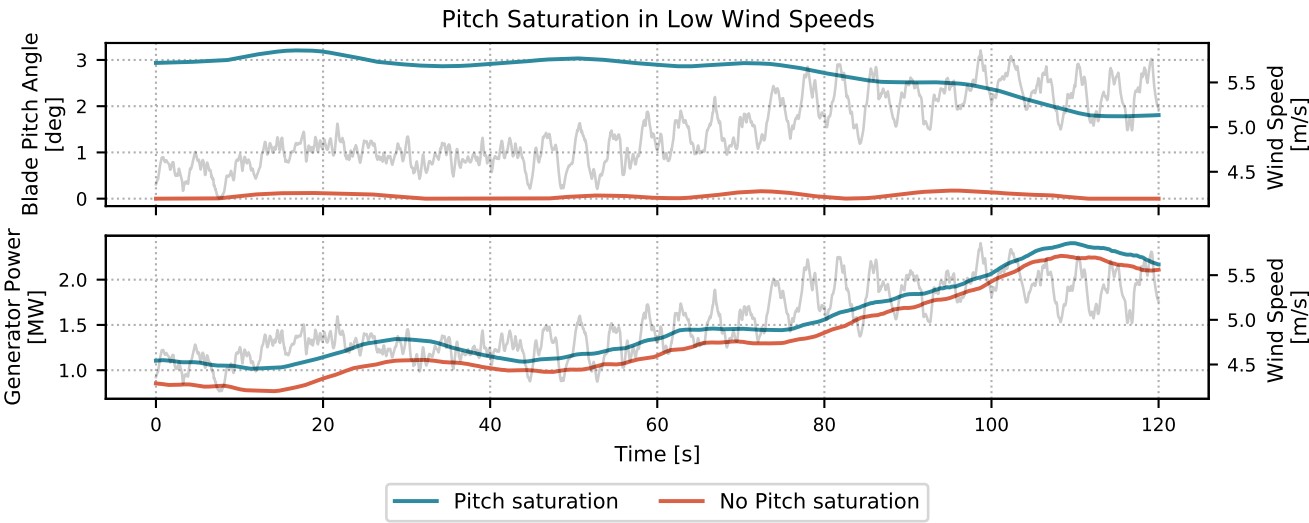

**Figure 10.** A sample 2-minute time series from a 10-minute turbulent simulation of the IEA 15-MW wind turbine showing how increased blade pitch angles can increase power output at low wind speed. The rotor-averaged wind speeds are plotted in grey in the background.

Figure 10 shows an example of how the power can be increased. A 10-minute simulation was run in OpenFAST for an inflow
wind with normal turbulence and a 5-$m/s$ average wind speed. With the pitch saturation module turned on, the blade pitch angle is changed based on the wind speed estimate and the pitch saturation lookup table, resulting in a slightly increased power output.

## 5.4   Shutdown

A simple shutdown routine is included in ROSCO for shutdown during high wind speed events. A first-order low-pass-filtered
blade pitch angle signal triggers turbine shutdown if it exceeds a certain threshold. If the shutdown is initiated, the blades are pitched to feather at their maximum pitch rate to slow down the rotor. If the TSR tracking torque controller is being used, $\omega_{\mathrm{ref},\tau}$ is set to the minimum rotor speed. This encourages the torque controller to help slow down the rotor initially. Once the blades are pitched such that very little lift is generated and the rotor is nearly stopped, the torque controller will saturate at zero in an unsuccessful attempt to speed up the rotor to the minimum rotor speed.
It is shown in Bottasso et al. (2014) that there are shutdown methods that could reduce possible design-driving loads during shutdown. Future work includes the investigation and inclusion of these methods in the ROSCO controller. Additionally, a number of events can trigger wind turbine shutdowns, such as generator overspeeds and yaw misalignment. Future work also includes the addition of improved shutdown event-detection and control methods for such cases.



## 5.5 Floating Offshore Wind Turbine Feedback

An additional control feedback term is included to account for FOWTs in a method referred to as "parallel compensation" (Van Der Veen et al., 2012). The tower-top acceleration is filtered, integrated, and multiplied times a proportional gain feedback, $k_{\beta_{\text{float}}}$. This modifies the blade pitch control signal to become:

$$d\beta = k_p d\omega_g + k_i \int_0^T d\omega_g dt + k_{\beta_{\text{float}}} d\dot{\phi}, \tag{47}$$

where $\phi$ is the tower-top pitch angle in the fore-aft direction. The block diagram in Figure 3 provides a visualization of how
this signal is implemented.

Although some research suggests the use of a platform pitch feedback signal for FOWT control (Fleming et al., 2014, 2016), the tower fore-aft signal is used in ROSCO so that the overall controller implementation can maintain the structure of the bladed-style communication interface (DNV-GL, 2018). A first-order high-pass filter combined with a second-order low-pass filter are used to filter the nacelle fore-aft rotational acceleration signal. The ROSCO toolbox generically places high- and
low-pass filter cutoff frequencies at $0.01\ rad/s$ and the platform's first fore-aft natural frequency, respectively. Additionally, a notch filter is used to remove the tower fore-aft frequency component of the feedback signal for the floating controller. A Bode diagram of the final form of this filter is shown in Figure 11 for the IEA 15-MW wind turbine on the University of Maine (UMaine) semisubmersible platform. After this nacelle fore-aft rotational acceleration signal is filtered and integrated, it is similar to a platform pitching velocity signal that is often used for FOWT control.

For consistency with the theme of the ROSCO tool set, a generic tuning process has been developed for this floating-feedback term. We start by defining the simplified second-order equation of tower-top motion as:

$$J_t d\ddot{\phi} + c_t d\dot{\phi} + k_t d\phi = l_t dT_r, \tag{48}$$

where $J_t$ is the total system inertia in the platform pitching direction, $c_t$ is a damping constant, $k_t$ is a restoring constant, $l_t$ is the tower height, and $T_r$ is the rotor thrust as defined by (45). The rotor effective wind speed is modified by tower motion such
that:

$$dv = dv_w - l_t d\dot{\phi}, \tag{49}$$

where $dv_w$ is the change in freestream wind speed. Similar to the derivation of (4), we can linearize (45) to be:

$$dT_r = \Pi_{\omega_g} d\omega_g + \Pi_\beta d\beta + \Pi_v dv, \tag{50}$$

where $\Pi_{\omega_g} = \frac{\partial T_r}{\partial \omega_g}$, $\Pi_\omega = \frac{\partial T_r}{\partial \beta}$, and $\Pi_\omega = \frac{\partial T_r}{\partial v}$.
For the controller, we have suggested that a proportional feedback term should be added to the standard blade pitch PI controller, such that the control input should be defined by (47). We also define the generator position as $\theta$ such that:

$$\dot{\theta} = \omega_g, \qquad \ddot{\theta} = \dot{\omega}_g. \tag{51}$$

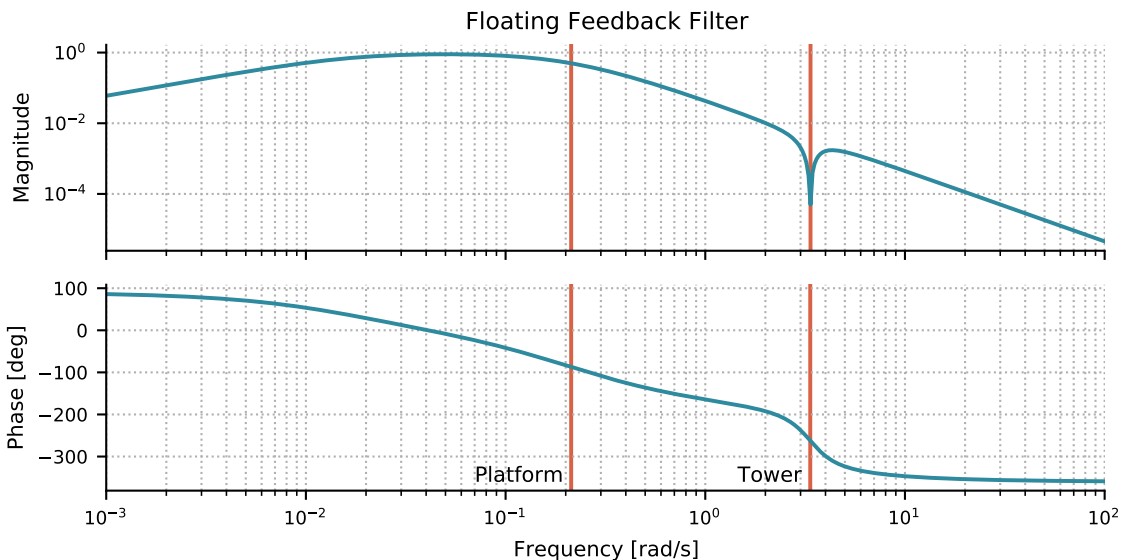

**Figure 11.** A Bode plot showing the filter used for the tower-top motion feedback signal for the IEA 15-MW turbine on the UMaine semisubmersible platform. The first platform and tower fore-aft natural frequencies are shown in red and labeled accordingly. These natural frequencies are inputs to the ROSCO tuning process to shape this filter.

After substituting (47) and (49)–(51) in equations (48) and (5), we arrive at the equations of motion for the closed-loop tower-top pitching (52) and rotor (53) dynamics:

$$d\ddot{\phi} = \frac{1}{J_t}\left[ -k_t d\phi + \left( -c_t - \Pi_v l_t^2 + \Pi_\beta l_t k_{\beta_{\text{float}}} \right)d\dot{\phi} + \Pi_\beta l_t k_i d\theta + \left( \Pi_\omega l_t + \Pi_\beta l_t k_p \right)d\dot{\theta} + \Pi_v l_t dv_w \right], \tag{52}$$

$$d\ddot{\theta} = \frac{N_g}{J}\left[ \left( \Gamma_\beta k_{\beta_{\text{float}}} - \Gamma_v l_t \right)d\dot{\phi} + \Gamma_\beta k_i d\theta + \left( \Gamma_{\omega_g} + \Gamma_\beta k_p \right)d\dot{\theta} + \Gamma_v dv_w - N_g d\tau_g \right]. \tag{53}$$

If we then convert equations (52) and (53) into the state-space form $\dot{x} = \boldsymbol{A}\boldsymbol{x} + \boldsymbol{B}\boldsymbol{u}$, where $\boldsymbol{u} = [dv_w \ \ d\tau_g]^T$ and $\boldsymbol{x} = [d\phi \ \ d\dot{\phi} \ \ d\theta \ \ d\dot{\theta}]^T$, we can define:

$$\boldsymbol{A} = \begin{bmatrix} 0 & 1 & 0 & 0 \\ -k_t & \left( -c_t - \Pi_v l_t^2 + \Pi_\beta l_t k_{\beta_{\text{float}}} \right) & \Pi_\beta l_t k_i & \left( \Pi_\omega l_t + \Pi_\beta l_t k_p \right) \\ 0 & 0 & 0 & 1 \\ 0 & \left( \Gamma_\beta k_{\beta_{\text{float}}} - \Gamma_v l_t \right) & \Gamma_\beta k_i & \left( \Gamma_{\omega_g} + \Gamma_\beta k_p \right) \end{bmatrix}, \quad \boldsymbol{B} = \begin{bmatrix} 0 & 0 \\ \Pi_v l_t & 0 \\ 0 & 0 \\ \Gamma_v & -N_g \end{bmatrix}. \tag{54}$$

Note that $A(4,2)$ is the state transition term from $d\dot{\phi}$ to $d\ddot{\theta}$. This suggests that if $\Gamma_\beta k_{\beta_{\text{float}}} - \Gamma_v l_t = 0$, then the tower-top pitching velocity will have no direct affect on the rotor acceleration. To attempt to achieve this, we define:

$$k_{\beta_{\text{float}}} = \frac{\Gamma_v}{\Gamma_\beta}l_t = \frac{\partial \tau_a}{\partial v}\left( \frac{\partial \tau_a}{\partial \beta} \right)^{-1} l_t \tag{55}$$

Including the additional parallel-compensation term $\beta_{\text{float}}$ reduces the need to detune the standard blade pitch PI controller because of the infamous negative-damping problem (Larsen and Hanson, 2007). If tuned appropriately, the parallel compen-





sation term can help negate the affect of tower-top motion and subsequent changes to the relative wind speed at the rotor. Additionally, combining the use of this $\beta_{\text{float}}$ feedback term with a peak shaving routine from Section 5.3.1 can reduce the tower-top pitching transients and further stabilize the system. Though the peak shaving does, theoretically, reduce power output near rated operation, the tower-top stabilization benefits generally outweigh the power losses because less fore-aft tower motion can lead to increased power output.

## 6   Example Results and Analysis

In this section, we use ROSCO v2.2.0 and present results from both land-based and floating wind turbine simulations. First, the ROSCO controller, as tuned by the ROSCO toolbox, is compared to the NREL 5-MW wind turbine controller using the land-based NREL 5-MW wind turbine. Then, results from the IEA 15-MW atop the UMaine semisubmersible floating wind turbine configuration (Allen et al., 2020) are shown to illustrate the effect of the FOWT $\beta_{\text{float}}$ control feedback loop. The subset

of the International Electrotechnical Commission (IEC) design load cases (DLCs) (DNV-GL, 2016) shown in Table 2 were run for both turbines to offer a comparison of controller performance.

**Table 2.** Simulated DLCs to showcase the difference in controller performance. The wind speeds were separated by increments of 2 $m/s$, with normal and extreme turbulence models defined by the IEC et al. (2006) standards. The sea-state-related inputs are relevant only for the FOWT simulations. Simulations were run for codirectional waves (i.e., wind and waves are aligned) to isolate the negative-damping phenomena that is generally of interest to the control designer.

| DLC | Wind Condition | Wind Speeds | Seeds | Waves |
|---|---|---|---|---|
| 1.1 | Normal turbulence | 3 to 25 m/s | 6 | Normal sea state, co-directional |
| 1.3 | Extreme turbulence | 3 to 25 m/s | 6 | Normal sea state, co-directional |

### 6.1   NREL 5-MW Land-Based Wind Turbine Results

The land-based NREL 5-MW turbine with the the NREL 5-MW wind turbine controller is compared to ROSCO with the ROSCO toolbox generic tuning values and ROSCO with minimal manual tuning and peak shaving implemented. As previ-

ously stated, there are only four *necessary* tuning parameters when tuning ROSCO using the generic ROSCO toolbox tuning functionalities. For the NREL 5-MW configuration, the choices for $\zeta_{\text{des}}$ and $\omega_{\text{des}}$ are shown in Table 3. The controller tuning parameters used for ROSCO's pitch controller in the results presented in this section were chosen to be the same as those used for scheduling the gains of the NREL 5-MW reference controller. ROSCO's TSR tracking torque controller was tuned manually so that below-rated simulations resulted in consistent TSR tracking.

The collective blade pitch controller gain schedules from the ROSCO tuning process are similar to those from the NREL 5-MW reference controller's tuning process, but they are not the same. The NREL 5-MW reference controller's gain schedule is based on the turbine's sensitivity of aerodynamic power to the collective blade pitch angle, a value that has generally been



**Table 3.** ROSCO controller tuning values for the collective blade pitch and TSR tracking torque controllers for the DLC simulations of the NREL 5-MW wind turbine.

|  | $\zeta_{\mathrm{des}}\ (-)$ | $\omega_{\mathrm{des}}\ (rad/s)$ |
|---|---|---|
| Collective blade pitch controller | 0.7 | 0.6 |
| Tip-speed ratio tracking torque controller | 0.7 | 0.15 |

found using more computationally expensive aeroelastic solvers, whereas the ROSCO tuning process depends directly on the $C_p$ surface.

High-level comparison results of the ROSCO controller compared to the NREL 5-MW reference controller are shown in Figure 12. Results are shown from both the generic ROSCO controller and a "tuned" ROSCO controller. For the tuned controller, all controller tuning input parameters were kept the same as for the generic ROSCO controller with the exception of modifying the set point smoother gains to be $k_{vs} = 5$ and $k_{pc} = 0.01$ and implementing the standard peak shaving routine. The results from the generic ROSCO controller are consistent with the expected steady-state operating points shown in Figure 2.

In the results from the generic ROSCO controller, there is a noticeable increase in rotor thrust compared to the NREL 5-MW reference controller. This is attributed to the difference in pitch controller gains, and it can be mitigated with slightly different controller tuning parameters. Also, note that the NREL 5-MW reference controller exhibits significantly higher TSRs because of the torque controller's linear transition between regions 1 and 2 (Jonkman et al., 2009) rather than the PI controller-based transition that is used in ROSCO. The ROSCO controller tracks the optimal TSR of 7.5 well during below-rated operation.

The power output and subsequent annual energy production from the ROSCO controller is 0.4% greater than that of the NREL 5-MW reference controller without peak shaving. This is consistent with observations by Holley et al. (1999) and Bossanyi (2000), who note that minor energy production gains to be achieved through an optimal TSR-tracking controller are possible, though at cost of more generator power fluctuations. The maximum standard deviations of the power output during DLC 1.1 are only 5% higher using ROSCO than when using the NREL 5-MW reference controller, so very minor

power production increases can be expected for the controller tuning used in this study. The annual energy production from the ROSCO controller is 1.7% less than that of the NREL 5-MW reference controller when the peak shaving routine is implemented in ROSCO. This is expected because rotor thrust reductions reduce the wind turbine's power output as well. Notably, the maximum rotor thrust seen is slightly higher than the expected maximum rotor thrust (see Figure 9). This is attributed to the imperfect nature of a peak shaving routine based on a wind speed estimate, and it could likely be improved through an observer-

based rotor thrust shaving algorithm or similar. The peak shaving also helps to both mitigate rotor thrusts introduced by the initial ROSCO tuning and reduce them further than those from the simulations using the NREL 5-MW reference controller.

### 6.2    IEA 15-MW on the UMaine Semisubmersible FOWT Results

We compare results from the IEA 15-MW wind turbine on a semisubmersible floating turbine platform for the ROSCO controller with and without the floating feedback and pitch saturation modules enabled (see Figure 3). Larsen and Hanson (2007)

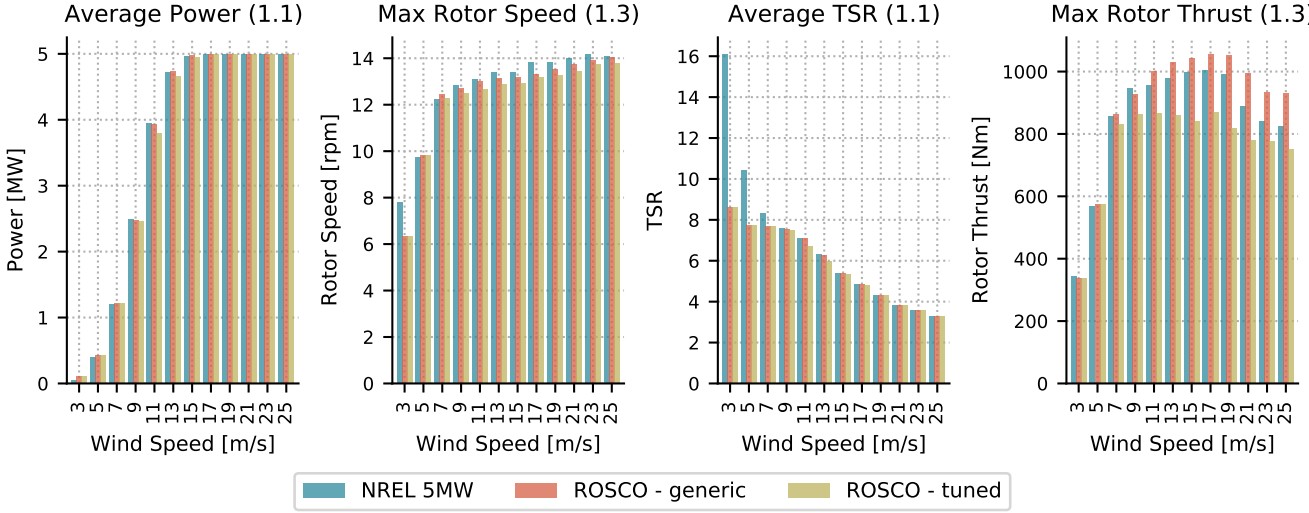

**Figure 12.** Results from simulations of the NREL 5-MW land-based wind turbine. Average power, maximum rotor speeds, average TSRs, and maximum rotor thrusts are shown for DLCs 1.1 and 1.3. The dynamic results from the ROSCO controller are comparable to those from the NREL 5-MW reference controller. "ROSCO - generic" refers to ROSCO with the most generic tuning methods, and "ROSCO - tuned" refers to ROSCO with tuned set point smoother gains and peak shaving implemented.

suggest that the tuned bandwidth of a rotor speed-regulating collective blade pitch controller should not be higher than the platform's first fore-aft eigenfrequency. This method of "derating" the blade pitch controller has traditionally been employed in baseline FOWT controllers, such as the NREL 5-MW OC3-and OC4-Hywind FOWT models (Jonkman, 2010; Robertson et al., 2014). Though derating the turbine can provide stable platform dynamics, generator overspeeds can generally be very high with this method of control. The standard distribution of the IEA 15-MW wind turbine on the UMaine semisubmersible

platform (IEA Wind Task 37, 2021) employs ROSCO, and the pitch controller bandwidth is $0.2\ rad/s$, which is already less than the UMaine semisubmersible's first platform eigenfrequency of ~ $0.21\ rad/s$. Also, the IEA 15-MW wind turbine uses constant generator torque control in above-rated operation. For this comparison, the floating feedback and pitch saturation modules are either enabled or disabled, and the rest of the controller is not changed at all.

Figure 13 presents results from the DLC 1.1 and 1.3 simulations. With the pitch saturation enabled, slight decreases in

power production are seen at near-rated wind speeds. This does, however, correlate to significant reductions of tower fore-aft bending moments and related loads near rated. The presented reduction in platform pitching motion of slightly more than 30% resulting from the floating-specific feedback term has a clear effect on the tower base bending moment as well. The above-rated maximum rotor speed is not expected to change significantly with and without the pitch saturation and floating controller because the standard pitch controller for rotor speed regulation is the same. Similarly, the average power output is expected to

drop when peak shaving is introduced, as shown in the bottom right plot of Figure 13.





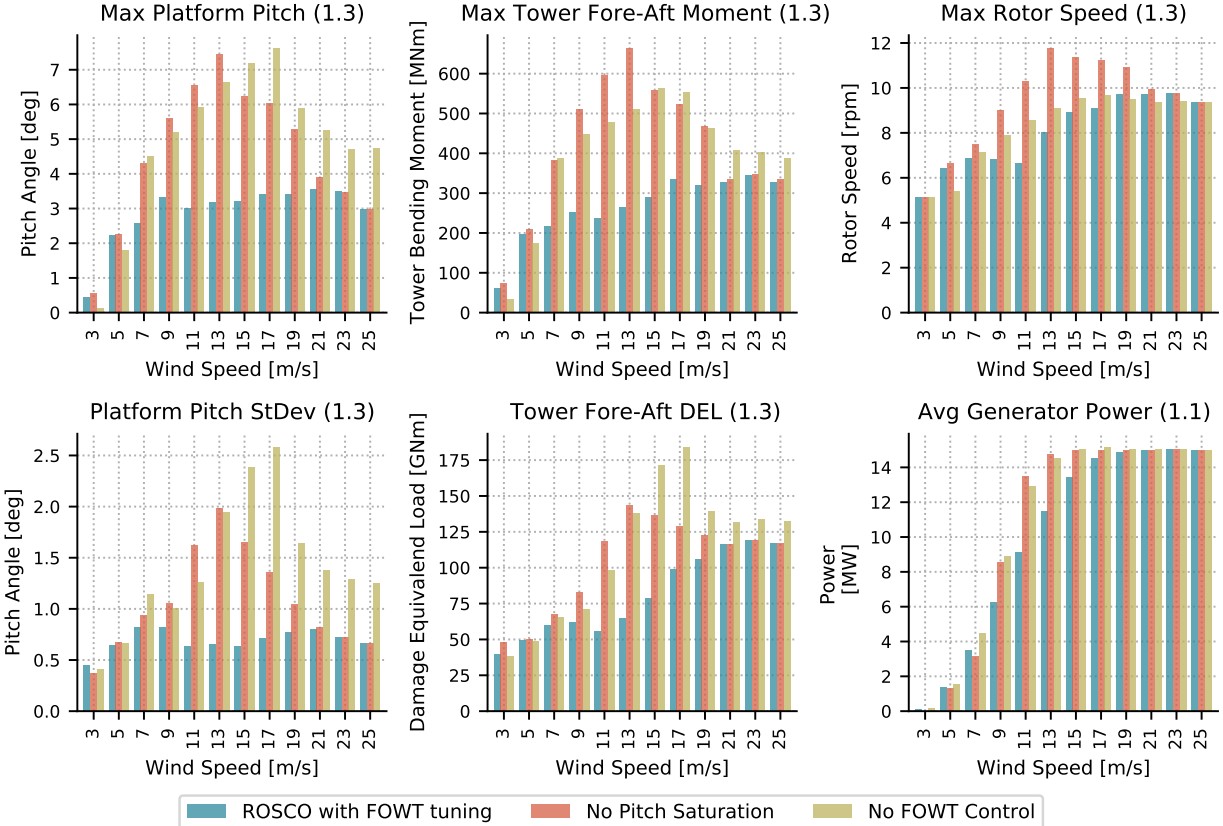

**Figure 13.** Results from DLC 1.1 and 1.3 simulations for the IEA 15-MW wind turbine on the UMaine semisubmersible platform. Output statistics from the platform pitch angles, tower base fore-aft bending moments, rotor speed, and generator power are shown. Simulations using the FOWT feedback loop ($\beta_{\text{float}}$) and pitch saturation are compared to those with the pitch saturation or FOWT feedback modules turned off.

The maximum platform pitch motions are reduced by approximately $47\%$, from 7.61 to 3.56 degrees, through use of the FOWT feedback term and pitch saturation routines. The annual energy production with complete FOWT tuning is $18.2\%$ less than that without the pitch saturation terms, though this is attributed to the near $20\%$ rotor overspeeds that are seen without peak shaving and fairly aggressive ($25\%$) peak shaving percentage. Lemmer et al. (2020) do a very good job of highlighting the trade-offs between focusing control actuation on regulating rotor speed versus mitigating platform pitch motions. In this work, no specific tuning of the floating feedback term or blade pitch controller gains was done outside of the methods presented in sections 5.5 and 2.5, so overall controller performance could certainly be tuned and improved.



# 7   Conclusions and Future Work

We have provided the research community with a reference open-source controller (ROSCO) for fixed and floating offshore wind turbines. The controller structure is similar to many controllers that are seen on industry turbines that function in the field. Generic tuning methods have been developed for the ROSCO controller and made available through the Python-based ROSCO toolbox. The tuning methods can be implemented easily by the interested non-controls engineer or in a completely automated fashion for use in optimization routines.

On the NREL 5-MW land-based wind turbine, we have shown that the generically tuned ROSCO controller performs comparably to the NREL 5-MW reference controller. We have also shown that, through a small amount of additional controller tuning, ROSCO's performance can be further improved to reduce rotor thrusts without significant reductions in power generation. We have also provided the foundations of more modern, industry standard control methods, such as peak shaving and wind speed estimation. A TSR tracking controller is shown to generate below-rated turbine power outputs that are consistent with the expectation based on the literature without significant power fluctuation or increased tower-base loads. No significant differences have been made to the implementation of the above-rated collective blade pitch controller, but automated tuning methods have been developed to easily generate pitch-dependent gain schedules for the blade pitch controller based on the $C_p$ surface. A set point smoother has been implemented to improve the transition between the primary blade pitch and generator torque controller regions. A Kalman filter-based wind speed estimator is implemented in ROSCO to enable the TSR tracking and blade pitch saturation capabilities. Pitch saturation routines have been implemented to improve power output in low wind speeds and reduce rotor thrust rated wind speeds. A simple shutdown logic has also been implemented to facilitate more realistic wind turbine testing for the IEC DLCs.

Additionally, a FOWT-specific feedback loop is included in ROSCO and has shown improvements over previously published open-source FOWT control methods. The combination of a FOWT-specific feedback loop and a peak shaving routine has been shown to significantly reduce platform pitch motions and maximum rotor speeds compared to a simple pitch controller with a bandwidth below the first fore-aft natural frequency of the platform. Notably, all the results shown for the IEA 15-MW on the UMaine semisubmersible are shown for the generalized ROSCO tuning values, and there is potential to further improve controller performance through additional fine-tuning of the input parameters to the ROSCO toolbox.

Several other capabilities are also being incorporated in ROSCO and the ROSCO toolbox generic tuning logic. These capabilities and tuning methods are currently being developed, some of which include individual pitch control, distributed aerodynamic control, and improved FOWT feedback term tuning methods. A number of specific improvements for the control methods discussed in this paper will also be implemented. These include additional functionalities for the blade pitch and torque controller, such as power-reference tracking control and tower resonance-avoidance methods. Improved shutdown and yaw methods are also being actively investigated.

Finally, the authors would like to reiterate that ROSCO and the ROSCO toolbox are open-source tools. We invite the research community to download, use, and contribute to these codes in whichever ways they see fit.





*Code availability.*

The ROSCO controller is available for download at:

$$\texttt{https://github.com/NREL/ROSCO.}$$

The ROSCO toolbox is available for download at:


$$\texttt{https://github.com/NREL/ROSCO\_toolbox.}$$

## Appendix A: Filters

Four filters are used in the ROSCO controller. For the filters presented, $\omega_f$ is a cornering frequency in $rad/s$, and $\zeta_f$ is a unitless damping ratio.

**First-order low-pass filter:**

$$F_{L1}(s) = \frac{\omega_f}{s + \omega_f} \tag{A1}$$

**Second-order low-pass filter:**

$$F_{L2}(s) = \frac{\omega_f^2}{s^2 + 2\zeta_f \omega_f s + \omega_f^2} \tag{A2}$$

**First-order high-pass filter:**

$$F_{H1} = \frac{s}{s + \omega_f} \tag{A3}$$

**Notch filter:**

$$F_N = \frac{s^2 + 2\omega_f \zeta_{f1} s + \omega_f^2}{s^2 + 2\omega_f \zeta_{f2} s + \omega_f^2} \tag{A4}$$

*Author contributions.* Nikhar Abbas developed the generic tuning processes, formalized the control concepts, and collected the theoretical foundations discussed in this article. He also contributed the majority of the code found in ROSCO and the ROSCO toolbox and established them as the tools that they are today. Nikhar Abbas was also the primary contributor to the writing of the article. Dan Zalkind contributed

to much of the testing and verification of the ROSCO toolchain. He has also served in an advisory role during the development of many of ROSCO's modules and reviewed this article multiple times before its submittal. Alan Wright contributed the original theoretical idea behind the tuning of the floating feedback term, served in an advisory role, and reviewed this article multiple times. Lucy Pao served in a supervisory role, guided the theoretical foundations discussed in this article, and reviewed this article extensively.

*Competing interests.* No competing interests are present.



*Disclaimer.* This work was authored in part by the National Renewable Energy Laboratory, operated by Alliance for Sustainable Energy, LLC, for the U.S. Department of Energy (DOE) under Contract No. DE-AC36-08GO28308. Funding provided by U.S. Department of Energy Office of Energy Efficiency and Wind Energy Technologies Office. The views expressed in the article do not necessarily represent the views of the DOE or the U.S. Government. The U.S. Government retains and the publisher, by accepting the article for publication, acknowledges that the U.S. Government retains a nonexclusive, paid-up, irrevocable, worldwide license to publish or reproduce the published form of this

work, or allow others to do so, for U.S. Government purposes.

*Acknowledgements.* The authors specifically thank Jan-Willem van Wingerden and Sebastiaan Mulders from the Delft University of Technology for their contributions to the Delft Research Controller, which provided much of the foundations upon which the ROSCO controller has been built.

Additionally, we thank the many researchers at the National Renewable Energy Laboratory who have put the entire ROSCO toolchain
through its paces in numerous applications. In particular, the authors thank Pietro Bortolotti and Evan Gaertner (now at Siemens Gamesa), who helped ROSCO through perhaps the most challenging software bugs. A special thanks to Paul Fleming as well for his initial inspiration on the structure of the ROSCO toolbox.



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
