# Peer review of "A Reference Open-Source Controller for Fixed and Floating Offshore Wind Turbines"

_Wind Energy Science, 2021_

## Author Comment (AC1)

First Author: Nikhar J. Abbas
Manuscript Title: A Reference Open-Source Controller for Fixed and Floating Offshore Wind
 Turbines
MS No.: wes-2021-19

**Author's Response to Reviewers**

We would like to thank the reviewers for taking the time to thoroughly review this manuscript. We have carefully considered all of their suggestions, and have attempted to acknowledge all suggested modifications, clarifications, and overall updates to the paper. The tables below directly address each of the reviewer comments and how we have updated the manuscript accordingly. All equation references correspond to the revised manuscript.

Please note that, outside of the changes directly addressed in this document, updates have been made to the results section of the manuscript in order to reflect the reviewers' suggested theoretical changes and more recent version of the ROSCO controller.

Additionally, a color-coded revised version is provided at the end of this document, which shows all of the changes that were made to the manuscript. All text in blue has been added to the manuscript, and the text in red has been removed. Figures that have received major revisions and updates are underlined in blue.

| Comments from Referee #1 | Authors' Response |
|---|---|
| Page 6, p. about Region 2.5: You mention that "due to the PI and controllers and the setpoint smoother, there is no specific range for Region 2.5". Since the PI controller and the setpoint smoother as implemented should not change the steady states, it should be always possible to calculate, at which wind speed the steady state of the rotor speed reaches rated rotor speed (depending on k of your k*Omega^2) and at which wind speed the static pitch is larger than the fine pitch. This should also allow to calculate the limits of region 2.5 even with the peak shaving. Please consider this. | Response
Thank you for pointing this out. We have tried to clarify the description of Region 2.5 in the manuscript.

Changes
We have added and also edited the text in the region 2.5 description in section 2.1. |
| Figure 2 left: TSR seems to continue to be constant at the end of region 2, but rated rotor speed is already reached (which is not possible). For the 5MW, rated rotor speed is also reached before rated power. It would be | Response
Thank you for noting this. The figures have been updated accordingly.
Changes |

| | |
|---|---|
| further helpful to add region 2.5 in the plot (see comment above). | Figure 2 has been updated to properly to show the tip speed ratio and to have a shaded area to denote region 2.5. |
| Section 2.5: The authors write that the integral gains are in general negative for standard horizontal-axis wind turbines. However, this depends on the definition of your speed error (reference – measurement or measurement-reference). From my perspective, the positive gains are more common (see e.g. Jonkman 2009). Please revise this part. | Response
Though it is true that Jonkman (and others) have denoted HAWT controller gains as positive, in ROSCO, the gains are negative because an increase of rotor speed results in a negative speed error, so an increase of collective blade pitch or generator torque is desired. This is consistent with others in the community, such as the Delft Research Controller. The manuscript has been updated to reflect *why* the gains are negative given our formulation. Section 2.5 now gives a more thorough formulation of the system used to tune the ROSCO controller. |
| | Changes
Changes have been made to section 2.5 to clarify our convention. Equations (12) and (13) have been added to clarify the form of the input and output perturbations. |
| Equation (15): Equation (15) is usually obtained setting the rotor motion from Equation (2) to zero and integrating the aerodynamic torque from Equation (3), see e.g. Bossanyi 2000. However, the efficiencies of the generator and gearbox are not part of Equation (2) and (3) of your paper. From my perspective, the efficiency of the gearbox should be part of Equation (2) and thus also (15), but the generator efficiency is only important to calculate the electrical power from the generator torque and thus should not be a part of Equation (2) and thus also not of Equation (15). Please make this part more consistent. | Response
Thank you for the detailed analysis of the equations – they have been updated accordingly. |
| | Changes
We have updated equations (2) and (15) (which is now equation (17) in the revised manuscript). |
| Equation (21): If "constant power" is used, one can also include the partial derivative of the generator torque with respect to the generator speed in Equation (5). Neglecting this usually causes a large deviation from my experience, also for the 5 MW reference | Response
Thank you for noting this missing piece of information. We have included a more detailed investigation into this term and updated the theory and results in the manuscript accordingly. |

| | |
|---|---|
| wind turbine. For the ROSCO controller and for the paper it would be nice, if you could include this part or provide some investigation that in your case this is neglectable | Changes
Equations (23) and (24) and their surrounding text have been added to Section 4 to clarify, theoretically, how constant power actuation can affect the second order linearized model (equation (5)) in above-rated operation. The results for the NREL 5MW wind turbine, which uses constant power operation, have been updated with the requisite theory included in the controller tuning process. |
| Section 5.5: Usually (in the Bladed interface), the tower top fore-aft acceleration is a translational degree-of-freedom and thus the integrated signal is the tower top fore-aft speed. This has been used in your reference (van der Veen 2012). However, you use the "tower top pitch angle" (i.e. rotational DOF), which is also possible (but much harder to measure/estimate in reality) and would provide similar results I assume. But since ROSCO is using the Bladed interface and aims to reflect the industrial state-of-the-art, please consider changing to the translational DOF. | Response
Thank you for noting this. The equations and results in the manuscript have been updated to reflect this suggestion. |
| | Changes
The formulation of the floating feedback gain in Section 5.5 has been updated to reflect the use of a tower top fore-aft translational acceleration signal. Additionally, the results for the IEA-15MW wind turbine have been updated with the updated controller. |
| Equation (5) etc.: Please use consider using $\delta$ instead of $d$ for $\beta$, $v$ etc. and introducing that $\Delta$ is the deviation from the steady state. Using simply $d$ might cause confusion with the operator "d". | Response
Thank you for pointing out the potential confusion here. The equations have been updated as suggested. |
| | Changes
$\Delta$ has been used throughout the manuscript as suggested in order to be clearer. |
| Equation (9) etc: Please consider that "d" is an operator and thus using $\mathrm{d}$ instead of simply $d$ would be more appropriate. | Response
Thank you for noting this, the equation has been updated. |
| | Changes
Many of the times that $d$ was used have now been converted to $\Delta$. $\mathrm{d}$ has been used in the necessary equations that remain (e.g. equation (9)). |

| | |
|---|---|
| Equation (6) etc: the tip speed ration might be good to introduce. And here, using $\partial$ as in Equation (4) would be more appropriate, since the tip speed ratio depends on both wind speed and generator speed. | Response
The tip speed ratio is introduced in equation (1). Thank you for noting the necessity for using $\partial$ - the equations have been updated accordingly. |
| | Changes
We have added a reference to the definition of the tip-speed ratio in (1) right after equations (6)-(8). $\partial$ is now used in all necessary equations. |
| Equation (12): The transfer function $H(s)$ connects the Laplace transform of the input to the one of the output. The Laplace transforms themselves do not depend on $s$. Thus, the fraction with d$\Omega_g(s)$ is a bit sloppy. Best might be to simply remove this and explain that the transfer function is obtained by using the Laplace transform and Equation (5) and (9). | Response
Thank you for noting this. The manuscript has been updated to be more theoretically rigorous. |
| | Changes
We have followed your suggestion and explicitly explained how the closed loop transfer function is obtained by taking the Laplace transforms of Equations (5) and (9) and combining them in a standard negative-feedback loop. |
| Figure 2 and Figure 5 caption, Appendix A etc.: Units are in non-italic in the rest of the paper (which makes sense, since they are not variables), but here you have $kNm$, $MNm$, etc.. Please consider changing them. | Response
Thank you for your attention to detail. The manuscript has been updated accordingly. |
| | Changes
All units have been changed to be non-italicized |
| Figure 3: setpoint smoother has more inputs than only the generator speed. | Response
Thank you for pointing this out, the figure has been updated. |
| | Changes
Figure 3 has been updated to more accurately show the setpoint smoother inputs. |
| l 297: "but the power is much more consistent" is not clear to me. Maybe just remove or add something to better explain it. Maybe you mean "more consistent compared to the constant torque case"? | Response
Thank you for pointing out this source of confusion – the manuscript has been updated for clarity. The subscript has also been fixed. |

| | |
|---|---|
| in Equation (20) you use "rat" as subscript, but in the rest of the paper "rated". Please consider to have this consistent. | Changes
The paper has been modified to specifically state that the power is more consistent in the constant power case than in the constant torque case, and the subscript has been fixed. |
| Section 3.1, last sentence: From my perspective, the proportional and integral gains for the torque PI controller are often chosen to be constant for simplicity, since applying Equation (13) usually does not provide significantly differences over the considered operation points. Please check if this could be also helpful here. The reason provided in the paper ("less erratic control actuation…") seems to be a bit vague for a Journal paper. | Response
Thank you for your comments on this. |
| | Changes
The phrasing here has been updated to note that fixing the controller gains simplifies the problem without negative effects. |
| l 377: Equation (17) should be included here since Equation (16) is TSR tracking only. | Response
Thanks for pointing this out, equation (17) (which is now equation (18) in the revised manuscript) has been included. |
| | Changes
The equation reference has been updated. |

| Comments from Referee #2 | Authors' Response |
|---|---|
| My main comment after going through the very extensive manuscript is: while undoubtedly the ROSCO controller and toolbox are a significant contribution to the community and represent a very large effort, the authors should also consider highlighting the scientific contributions of the paper, avoiding reading the article as a "ROSCO user manual". It seems that most of the effort has been towards implementing well known methods and approaches, and automatizing some of the processes, which is of course a great effort that merits appreciation, however it does not automatically and necessarily entail publication in a scientific journal. | Response
Thank you for commenting on this – we certainly see how there may be some confusion as to whether or not this should have been published as a user manual. We believe this manuscript is much more than a user manual, and we have updated the manuscript and tried to improve clarity as to what the scientific contributions of this work are, and why this work merits publication in a scientific journal.

Additionally, we would like to note that this manuscript provides very little information as to how to actually use and implement the ROSCO tools. The paper itself focuses on the mathematical methods used for tuning and implementation of the controller, and how they are applied within the context of wind turbines. Though some of the methods and approaches have been discussed elsewhere in the literature (as cited), to the author's knowledge, there is no other publication that provides the theoretical and mathematical detail on automated controller tuning and implementation methods as completely as this one does. |
|  | Changes
Changes have been made to the introduction to further highlight what we believe to be the scientific contributions of this work. |

[revised manuscript text omitted]

$$\underline{d}\underline{\Delta}\beta = k_p\underline{d}\underline{\Delta}\omega_g + k_i \int_0^T \underline{d}\underline{\Delta}\omega_g dt + k_{\beta_{\text{float}}}\underline{d}\dot{\underline{\phi}}\underline{\Delta}\dot{x}_t, \tag{52}$$

where $\dot{\phi}$ $\underline{\dot{x}_t}$ is the tower-top  position in the fore-aft direction. The block diagram in Figure 3 provides a visualization of how this signal is implemented.
490  Although some research suggests the use of a platform pitch feedback signal for FOWT control (Fleming et al., 2014, 2016), the  nacelle fore-aft signal is used in ROSCO so that the overall controller implementation can maintain the structure

of the bladed-style communication interface (DNV-GL, 2018). A first-order high-pass filter combined with a second-order low-pass filter are used to filter the nacelle fore-aft  acceleration signal. The ROSCO toolbox generically places high- and low-pass filter cutoff frequencies at  0.01 rad/s and the platform's first fore-aft natural frequency, respectively.

495 Additionally, a notch filter is used to remove the tower fore-aft frequency component of the feedback signal for the floating controller. A Bode diagram of the final form of this filter is shown in Figure 12 for the IEA 15-MW wind turbine on the University of Maine (UMaine) semisubmersible platform (Allen et al., 2020). After this nacelle fore-aft  acceleration signal is filtered and integrated, it is similar to a platform pitching velocity signal that is often used for FOWT control.

[Figure]

**Figure 12.** A Bode plot showing the filter used for the tower-top motion feedback signal for the IEA 15-MW turbine on the UMaine semisubmersible platform. The first platform and tower fore-aft natural frequencies are shown in red and labeled accordingly. These natural frequencies are inputs to the ROSCO tuning process to shape this filter.

For consistency with the theme of the ROSCO tool set, a generic tuning process has been developed for this floating-feedback

500 term. We start by defining the simplified second-order equation of tower-top motion as:

$$J_t \underline{d\ddot{\phi}}\ddot{x}_t + c_t \underline{d\dot{\phi}}\dot{x}_t + k_t \underline{d\phi} = \underline{lx_t}\underline{dT} \equiv \underline{T}_r, \tag{53}$$

where $J_t$ is the total system inertia in the platform pitching direction, $c_t$ is a damping constant, $k_t$ is a restoring constant,  and $T_r$ is the rotor thrust as defined by (50). The rotor effective wind speed is modified by tower motion such that:

505 $$\underline{dv}\Delta v = \underline{dv}\Delta v_w - \underline{l}\Delta \dot{x}_t \underline{d\dot{\phi}}, \
[revised manuscript text omitted]

---

## Referee Report (RR1)

**Review of Manuscript Submitted to Wind Energy Science**

Subject: Second round of review based on the revised paper
Title: A Reference Open-Source Controller for Fixed and Floating Offshore Wind Turbines
Authors: Nikhar J. Abbas, Daniel S. Zalkind, Lucy Pao, and Alan Wright

The revised manuscript under this second round of review is in general well written and has addressed most of the comments proposed during the first round of the review. Therefore, my recommendation is to accept the paper with minor revisions. Few additional comments to further increase the quality of the paper are listed below:

- Line 17: As stated in the revised manuscript, the proposed open-source controller reduces the maximum thrust by over 10%. It also reduces the maximum platform pitch angle by approximately 20% when using the platform feedback routine instead of a more traditional low-bandwidth controller. These numbers are shown to convince the reader of the merit of the implemented algorithms, and I agree they look promising. However, I am missing a comparative analysis of the wind turbine power output and in particular the annual energy production. If a reduction in the turbine thrust happens at the same time with a reduction of the annual energy production, then it is questionable if the proposed change has a positive overall impact. I suggest that the authors consider adding a simple calculation to evaluate the annual energy production for a given site of their choice and compare the results. If a reduction in the maximum thrust happens at the same time with an increase of annual energy production or at least the same annual energy production, then the authors can fairly conclude that the proposed open-source controller is effective and better than the existing controllers.

- It would be nice if the authors could provide a detailed view of the changes to the wind turbine loading for different DLCs, and at critical locations such as blade root, high and low speed shafts, and tower top and bottom.

- Section 2.1 and conclusion: The controller design process of industrial wind turbines is more detailed than the simplified design process explained in this paper. I suggest that the authors clearly explain that their approach is suitable for an automated design of mainly region 2 and 3 based on fundamental control design principles available in the public literature. Having statements like line 606 (*"The controller structure is similar to many controllers that are seen on industry turbines that function in the field."*) is not realistic and does not add any value to the paper. Real wind turbines operating in the field are more sophisticated in structure and algorithm than the proposed controller, and what is known to the public research community.

---

## Author Response (AR2)

First Author: Nikhar J. Abbas
Manuscript Title: A Reference Open-Source Controller for Fixed and Floating Offshore Wind
                   Turbines
MS No.: wes-2021-19

**Author's Response to Reviewers**

We would like to thank the reviewer for taking the time to review our revised manuscript. We have carefully considered all of their suggestions, and have attempted to acknowledge all suggested modifications, clarifications, and overall updates to the paper. The tables below directly addresses each of the reviewer comments and how we have updated the manuscript accordingly. All equation references correspond to the revised manuscript.

In addition to the changes directly addressed in this document, a few minor updates have also been made to our manuscript to improve clarity of some ideas.

Further, a color-coded revised version is provided at the end of this document, which shows all of the changes that were made to the manuscript. All text in blue has been added to the manuscript, and the text in red has been removed. Figure 15 has also been added for this revision.

| Comments from Referee #1 | Authors' Response |
|---|---|
| Line 17: As stated in the revised manuscript, the proposed open-source controller reduces the maximum thrust by over 10%. It also reduces the maximum platform pitch angle by approximately 20% when using the platform feedback routine instead of a more traditional low-bandwidth controller. These numbers are shown to convince the reader of the merit of the implemented algorithms, and I agree they look promising. However, I am missing a comparative analysis of the wind turbine power output and in particular the annual energy production. If a reduction in the turbine thrust happens at the same time with a reduction of the annual energy production, then it is questionable if the proposed change has a positive overall impact. I suggest that the authors consider adding a simple calculation to evaluate the annual energy production for a given site of their choice and compare the results. If a reduction in the maximum thrust happens at the same time with an increase of annual energy production or at least the same | Response
Thank you for this feedback. A more detailed analysis of the IEA-15MW wind turbine has been included to better highlight the tradeoffs of using different outputs of various controller implementations on the IEA-15MW wind turbine. |
|  | Changes
An AEP comparison is now provided in Figure 15 and is discussed in the surrounding text. |

| | |
|---|---|
| annual energy production, then the authors can fairly conclude that the proposed open-source controller is effective and better than the existing controllers. | |
| It would be nice if the authors could provide a detailed view of the changes to the wind turbine loading for different DLCs, and at critical locations such as blade root, high and low speed shafts, and tower top and bottom. | Response
We appreciate this suggestion. Some more details of the DLC results are provided. |
| | Changes
An analysis of changes for the IEA-15MW with different controller configurations is provided in Section 6.2. Figure 15 has also been added to show the changes graphically. |
| Section 2.1 and conclusion: The controller design process of industrial wind turbines is more detailed than the simplified design process explained in this paper. I suggest that the authors clearly explain that their approach is suitable for an automated design of mainly region 2 and 3 based on fundamental control design principles available in the public literature. Having statements like line 606 (*"The controller structure is similar to many controllers that are seen on industry turbines that function in the field.")* is not realistic and does not add any value to the paper. Real wind turbines operating in the field are more sophisticated in structure and algorithm than the proposed controller, and what is known to the public research community. | Response
Thank you for offering this feedback, it is well received. We have removed some statements and clauses that suggest that this controller might be very representative of an industry controller. |
| | Changes
The statement in like 606 has been removed, and the abstract and Section 2 have been updated for clarity. |